# zGrad is a nanobody-based degron system that inactivates proteins in zebrafish

**Naoya Yamaguchi, Tugba Colak-Champollion, Holger Knaut***

Skirball Institute of Biomolecular Medicine, New York University School of Medicine, New York, United States

**Abstract** The analysis of protein function is essential to modern biology. While protein function has mostly been studied through gene or RNA interference, more recent approaches to degrade proteins directly have been developed. Here, we adapted the anti-GFP nanobody-based system deGradFP from flies to zebrafish. We named this system zGrad and show that zGrad efficiently degrades transmembrane, cytosolic and nuclear GFP-tagged proteins in zebrafish in an inducible and reversible manner. Using tissue-specific and inducible promoters in combination with functional GFP-fusion proteins, we demonstrate that zGrad can inactivate transmembrane and cytosolic proteins globally, locally and temporally with different consequences. Global protein depletion results in phenotypes similar to loss of gene activity, while local and temporal protein inactivation yields more restricted and novel phenotypes. Thus, zGrad is a versatile tool to study the spatial and temporal requirement of proteins in zebrafish.
DOI: https://doi.org/10.7554/eLife.43125.001

## Introduction

The study of the consequences of the loss of gene function is a central technique in biology. In principle, loss of gene function can be achieved through gene, mRNA or protein inactivation. While there are many techniques available to inactivate genes and mRNA globally, spatially and temporally (*Housden et al., 2017*), methods to reduce protein function in the same manner are more limited. One strategy to inactivate a specific protein is to fuse it to a degron, a protein domain that targets its fusion partner for degradation. Degrons are recognized by adapter proteins. These adapter proteins target the degron together with its fusion partner to the E3 ubiquitin ligase complex for degradation by the proteasome. By manipulating degrons with temperature, light, small molecules or another protein, protein-degron fusions can also be degraded in a controlled manner (*Natsume and Kanemaki, 2017*). While different versions of these approaches have been adapted to zebrafish, it is mostly unclear whether they degrade the protein-degron efficiently enough to cause a phenotype (*Bonger et al., 2014*; *Shin et al., 2015*; *Daniel et al., 2018*; *Neklesa et al., 2011*).

GFP-based fluorescent proteins (FPs) are convenient degrons because they are often used to tag proteins and their degradation can be easily monitored by light microscopy. deGradFP, a system developed in flies, depletes FP-tagged proteins (*Caussinus et al., 2011*; *Nagarkar-Jaiswal et al., 2015*). It relies on the expression of an anti-GFP nanobody/F-box fusion protein. This fusion protein recruits FP-tagged proteins to the SKP1- CUL1-F-box (SCF) E3 ubiquitin ligase complex, leading to its ubiquitylation and proteasome-mediated degradation in 2-3 hr. Slight variations of this system have been used to deplete FP-tagged proteins in different tissues in nematodes (*Wang et al., 2017*), flies (*Brauchle et al., 2014*) and in zebrafish nuclei (*Shin et al., 2015*).

Here, we surveyed different degron systems in zebrafish and identified an anti-GFP nanobody/F-box fusion protein, which we named zGrad, that efficiently degrades FP-tagged proteins in

**\*For correspondence:**
Holger.Knaut@med.nyu.edu

**Competing interests:** The authors declare that no competing interests exist.

zebrafish. We used this system to show that zGrad depletes FP-tagged proteins within 30 to 150 min depending on the subcellular localization and the nature of the tagged protein. We found that zGrad-mediated protein degradation recapitulates genetic loss-of-function phenotypes and can uncover severe maternal and maternal-zygotic phenotypes. When we induced zGrad from a heat-shock-inducible promoter, we found temporal degradation of FP-tagged proteins with acute and reversible loss-of-protein function phenotypes. Lastly, we expressed zGrad from a tissue-specific promoter and found degradation of FP-tagged proteins exposing a phenotype similar to but less severe than the genetic loss-of-function phenotype. These observations indicate that zGrad is a versatile tool to rapidly and reversibly deplete FP-tagged proteins with temporal or spatial control in zebrafish.

## Results

### zGrad efficiently degrades GFP-tagged proteins in zebrafish

To establish a method to reduce the activity of proteins in zebrafish, we tested several degron-based protein degradation systems. First, we tested the auxin-inducible degron (AID) system (*Natsume et al., 2016*; *Nishimura et al., 2009*; *Holland et al., 2012*; *Gu et al., 2018*). This system uses the plant F box transport inhibitor response 1 (TIR1) protein to recruit proteins tagged with the small AID degron to the SCF E3 ubiquitin ligase complex in an auxin-dependent manner (*Figure 1A*). We co-injected one-cell stage embryos with *sfGFP-mAID* mRNA and varying amounts of *OsTIR1-mCherry* mRNA and treated them for 3 hr with or without the auxin analog indole-3-acetic acid (IAA). We found that expression of OsTIR1-mCherry induced degradation of sfGFP-mAID in a dose-dependent manner irrespective of IAA (*Figure 1D*, *Figure 1—figure supplement 1A*). We also found that 50 pg/embryo or higher concentrations of *OsTIR1-mCherry* mRNA led to deformed and dying embryos (*Figure 1—figure supplement 1B,C*). Thus, low levels of OsTIR1 induce degradation of AID-tagged proteins, higher levels of OsTIR1 are toxic and OsTIR degrades AID-tagged proteins in the absence of auxin, suggesting that, in its current form, the AID system is not suitable to degrade proteins in zebrafish.

Second, we tested the ZIF-1/ZF1 tag degradation system. This system uses the *C. elegans* SOCS-box adaptor protein ZIF-1 which binds to the zinc-finger domain ZF1 to recruit ZF1-containing protein to the ECS (Elongin-C, Cul2, SOCS-box family) E3 ubiquitin ligase complex for proteasomal destruction (*Figure 1A*) (*Armenti et al., 2014*). We co-injected one-cell stage embryos with *sfGFP-ZF1* mRNA, *zif-1* mRNA and *mScarlet-V5* mRNA and assessed sfGFP-ZF1 degradation 9 hr later at the epiboly stage. The sequences of the ZF1 tag and ZIF-1 were codon optimized for zebrafish and mScarlet-V5 served as an internal standard. Using the ratio of sfGFP to mScarlet fluorescence intensity as a measure, we found that the ZIF-1/ZF1 tag degradation only reduced the levels of sfGFP-ZF1 by 17% (*Figure 1E,H*), indicating that the ZIF-1/ZF1 tag degradation system does not efficiently degrade ZF1-tagged proteins in zebrafish.

Third, we tested the Ab-SPOP/FP-tag degradation system. This system uses the CULLIN-binding domain from the human Speckle Type BTB/POZ Protein (SPOP) adaptor protein fused to the single domain anti-GFP nanobody vhhGFP4 (*Figure 1A*) (*Shin et al., 2015*). The Ab-SPOP hybrid adaptor protein targets nuclear but not cytoplasmic FP-tagged proteins for degradation. We co-injected one-cell stage embryos with *sfGFP-ZF1* mRNA, *Ab-SPOP* mRNA or *Abmut-SPOP* mRNA and *mScarlet-V5* mRNA and assessed sfGFP-ZF1 degradation 9 hr later at the epiboly stage. Abmut-SPOP is a negative control and does not bind FPs because the GFP-binding domain in the nanobody is deleted (*Shin et al., 2015*). We assessed the degree of GFP degradation as the ratio of sfGFP to mScarlet fluorescence intensity in the cytoplasm and the nucleus. Consistent with the initial description (*Shin et al., 2015*), Ab-SPOP efficiently degraded nuclear sfGFP-ZF1 (70% reduction, *Figure 1F,H*, *Figure 1—figure supplement 2*) but not cytoplasmic sfGFP-ZF1 (13% reduction, *Figure 1F,H*, *Figure 1—figure supplement 2*). Abmut-SPOP did not cause any detectable sfGFP-ZF1 degradation (*Figure 1F*, *Figure 1—figure supplement 2*). As reported previously (*Shin et al., 2015*), this confirms that the Ab-SPOP/FP-tag degradation system degrades nuclear proteins efficiently but is not suitable for the degradation of non-nuclear proteins in zebrafish.

Fourth, we tested the degrade Green Fluorescent Protein (deGradFP) system. This system uses the F-box domain from the *Drosophila* Slimb adaptor protein fused to the anti-GFP nanobody

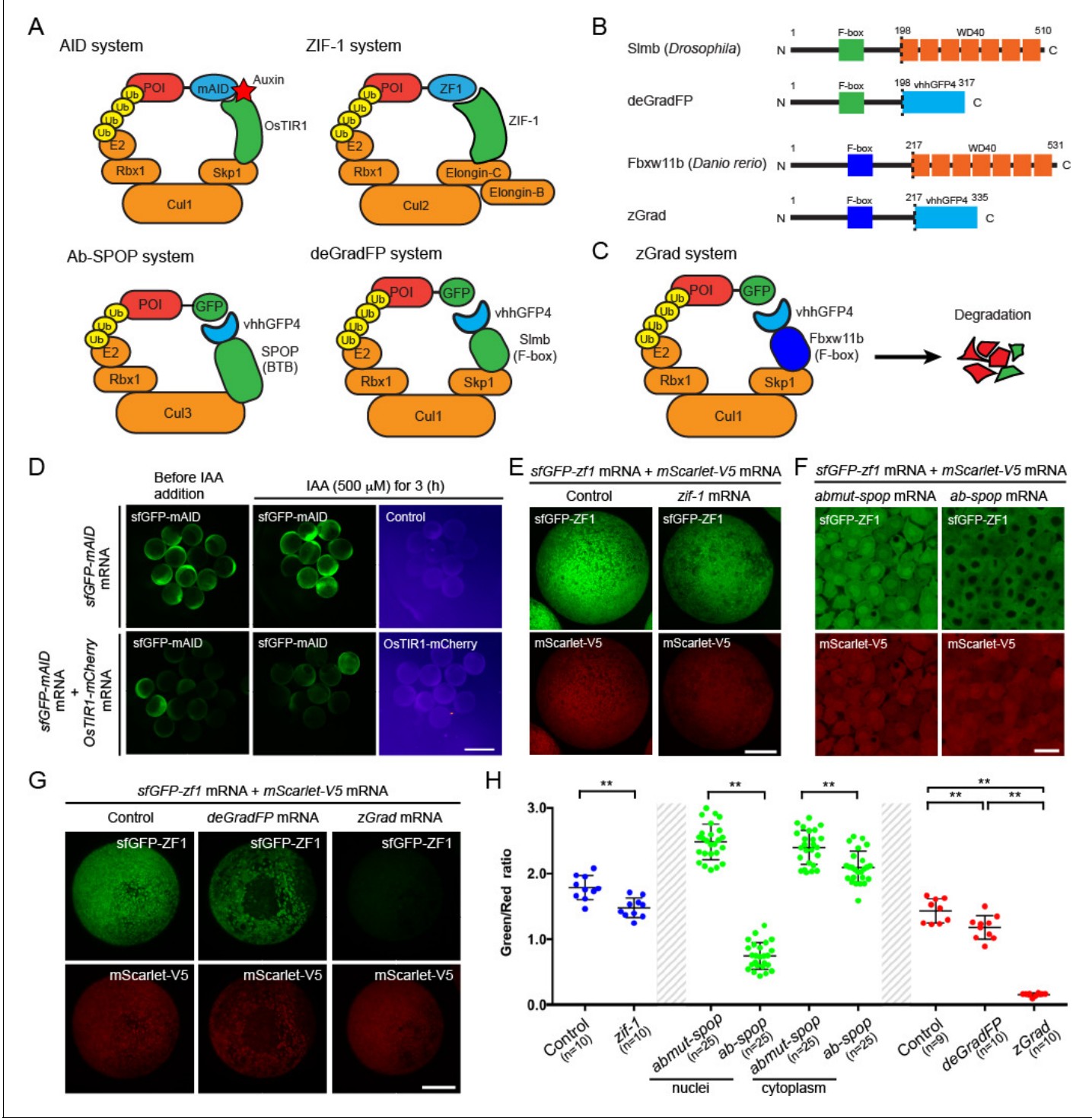

**Figure 1.** zGrad degrades GFP-tagged proteins in zebrafish. (**A**) Overview of degron-based protein degradation systems. POI: protein of interest. (**B**) Comparison of deGradFP and zGrad fusion proteins. (**C**) Schematic of zGrad-mediated target protein degradation. (**D**) Representative images of embryos injected with *sfGFP-mAID* mRNA (210 pg) only (top) or with *sfGFP-mAID* mRNA (210 pg) and *OsTIR1-mCherry* mRNA (42 pg) (bottom) before (left, 8 hpf) and after IAA (500 µM) induction for 3 hr (middle and right, 11 hpf). Red channel is shown as a fire map (right). Scale bar: 1 mm. (**E**) Representative images of embryos injected with *sfGFP-ZF1* mRNA and *mScarlet-V5* mRNA only (left) or with *sfGFP-ZF1* mRNA and *mScarlet-V5* mRNA and *zif-1* mRNA (right) at 9 hpf. Scale bar: 200 µm. Note that mScarlet-V5 fluorescence served as an internal control. (**F**) Single-plane confocal images of cells in embryos injected with *sfGFP-ZF1* mRNA, *mScarlet-V5* mRNA and *abmut-spop* mRNA (left) or *ab-spop* mRNA (right) at 9 hpf. Scale bar: 20 µm. (**G**) Representative images of embryos injected with *sfGFP-ZF1* mRNA and *mScarlet-V5* mRNA (left) or with *sfGFP-ZF1* mRNA and *mScarlet-V5* mRNA

*Figure 1 continued on next page*

*Figure 1 continued*

and *deGradFP* mRNA (middle) or *zGrad* mRNA (right) at 9 hpf. Scale bar: 200 µm. (**H**) Quantification of control and Zif-1-mediated sfGFP-ZF1 degradation shown in E (blue), Abmut-SPOP control and Ab-SPOP-mediated sfGFP-ZF1 degradation shown in F (green), and deGradFP-mediated and zGrad-mediated sfGFP-ZF1 degradation shown in G (red). Mean, SD and n are indicated. \*\*p<0.01. .

DOI: https://doi.org/10.7554/eLife.43125.002

The following figure supplements are available for figure 1:

**Figure supplement 1.** Characterization of the AID system in zebrafish.

DOI: https://doi.org/10.7554/eLife.43125.003

**Figure supplement 2.** Characterization of the Ab-SPOP system in zebrafish.

DOI: https://doi.org/10.7554/eLife.43125.004

**Figure supplement 3.** Onset of zGrad-mediated sfGFP-ZF1 degradation in early embryos.

DOI: https://doi.org/10.7554/eLife.43125.005

vhhGFP4 to target FP-tagged proteins for degradation (*Figure 1A and B* and *Caussinus et al., 2011*). We co-injected one-cell stage embryos with *sfGFP-ZF1* mRNA, *deGradFP* mRNA and *mScarlet-V5* mRNA and assessed sfGFP-ZF1 degradation 9 hr later at the epiboly stage. As above, mScarlet-V5 served as an internal standard and GFP degradation was assessed as the ratio of sfGFP to mScarlet fluorescence intensity. We found that deGradFP reduced sfGFP-ZF1 only slightly (19% reduction, *Figure 1G,H*). Since the fusion of the anti-GFP nanobody to the Cullin-binding domain from SPOP resulted in efficient albeit only nuclear degradation of GFP, we reasoned that the anti-GFP nanobody recognizes GFP-tagged proteins but that the Slimb F-box domain is not efficiently recruited to the E3 ligase complex in zebrafish. We therefore replaced the Slimb F-box domain in deGradFP with the homologuous F-box domain from zebrafish, reasoning that this should result in more efficient GFP degradation in zebrafish. Based on sequence homology, we identified the zebrafish *F-box and WD repeat domain containing 11b* (*fbxw11b*) gene as the *Drosophila slmb* orthologue. We then replaced the *Drosophila* F-box domain from Slimb in deGradFP with the zebrafish F-box domain from Fbxw11b (*Figure 1B,C*). We named this hybrid adaptor protein zGrad for zebrafish deGradFP. To test whether zGrad degrades GFP-tagged proteins, we co-injected one-cell stage embryos with *sfGFP-ZF1* mRNA, *zGrad* mRNA and *mScarlet-V5* mRNA and assessed sfGFP-ZF1 degradation 9 hr later at the epiboly stage using the ratio of sfGFP to mScarlet fluorescence intensity. In contrast to deGradFP, which depleted 19% of sfGFP-ZF1, zGrad depleted sfGFP-ZF1 in both the cytoplasm and the nucleus by 89% (*Figure 1G,H*). Importantly, we observed degradation of sfGFP-ZF1 by zGrad already at 2.5 hpf (*Figure 1—figure supplement 3*). Together, this indicates that zGrad efficiently targets tagged GFP in the cytoplasm for degradation and should be suitable for assessing protein function in zebrafish.

## zGrad degrades nuclear, transmembrane and cytoplasmic FP-tagged proteins

Next, we asked whether zGrad degrades GFP-tagged proteins in different cellular compartments using a transgenic line that expresses zGrad from a heat-shock-inducible promoter in the embryo (*hsp70l:zGrad*).

First, we tested whether zGrad degrades the nuclear protein Histone 2A (H2A). We generated *hsp70l:zGrad* embryos that also expressed H2A-EGFP and H2A-mCherry from two identical 69 kb genomic fragments spanning the *cxcr4b* locus (*cxcr4b:H2A-GFP; cxcr4b:H2A-mCherry*). Among other tissues, the *cxcr4b* promoter drives expression in the somites and the posterior lateral line primordium (primordium) (*Figure 2—figure supplement 1*) (*Chong et al., 2001*). Such embryos were heat shocked at 30 hpf for 1 hr and imaged over 9.5 hr. Compared to control embryos that did not carry the *hsp70l:zGrad* transgene, H2A-EGFP degradation was discernible in zGrad-expressing embryos within 2-3 hr post heat shock in all tissues that expressed nuclear EGFP from the *cxcr4b* promoter (skin, pronephros, somites, neural tube and primordium, *Figure 2A–E*, *Figure 2—Video 1*). We quantified H2A-EGFP levels in heat-shocked control embryos and heat-shocked *hsp70l:zGrad* embryos using the fluorescence intensity of H2A-mCherry as a reference. Since H2A-mCherry was expressed from the same promoter as H2A-EGFP and since H2A-mCherry is not recognized by the anti-GFP nanobody and should not be subjected to zGrad-mediated protein degradation (*Caussinus et al., 2011*), comparing the ratio of H2A-mCherry expression levels to H2A-EGFP should

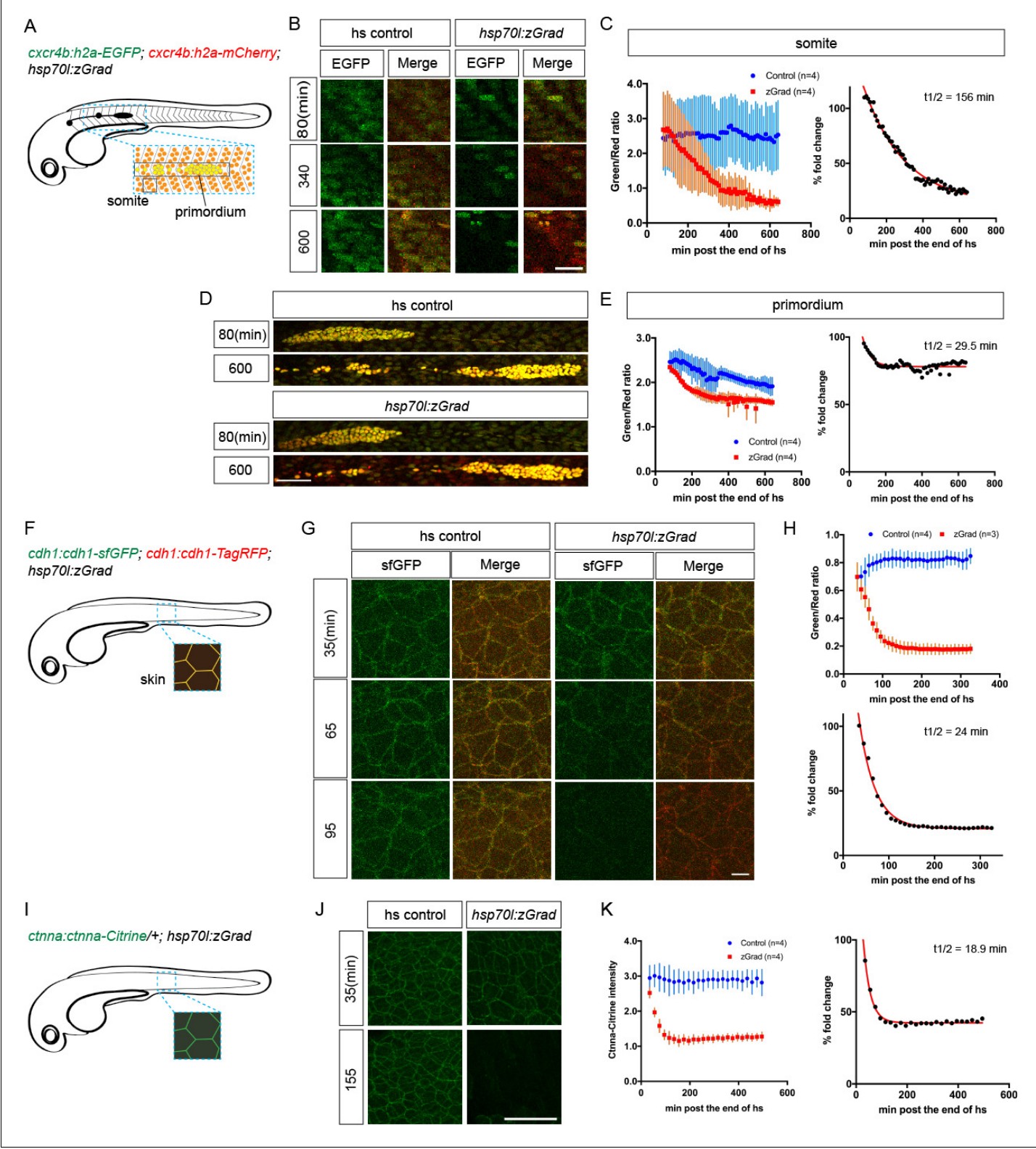

**Figure 2.** zGrad degrades nuclear, transmembrane and cytoplasmic proteins. (**A**) Schematic of strategy to assess zGrad-mediated H2A-EGFP degradation in the somites and the primordium. (**B**) Maximum-projected confocal images of nuclei in the somites in heat-shocked *cxcr4b:H2A-EGFP*; *cxcr4b:H2A-mCherry* embryos transgenic for *hsp70l:zGrad* (right) or not (left) at indicated time in min after the end of heat shock (29–30 hpf). Scale bar: 20 μm. (**C**) Left, quantification of H2A-EGFP-to-H2A-mCherry ratios in the somites of control (blue) and zGrad-expressing embryos (red) after the end of heat shock in min. Mean and SD are indicated. Right, H2A-EGFP-to-H2A-mCherry ratio of zGrad-expressing embryos normalized to control embryos

*Figure 2 continued on next page*

*Figure 2 continued*

(black dots) and fitted to a one-exponential decay model (red). (**D**) Maximum-projected confocal images of primordium nuclei in heat-shocked *cxcr4b: H2A-EGFP; cxcr4b:H2A-mCherry* embryos transgenic for *hsp70l:zGrad* (bottom) or not (top) at indicated time in min after the end of heat shock (29–30 hpf). Scale bar: 50 μm. (**E**) Left, quantification of H2A-EGFP-to-H2A-mCherry ratios in the primordia of control (blue) and zGrad-expressing embryos (red) after the end of heat shock in min. Mean and SD are indicated. Right, H2A-EGFP-to-H2A-mCherry ratio of zGrad-expressing embryos normalized to control embryos (black dots) and fit to a one-exponential decay model (red). (**F**) Schematic of strategy to assess zGrad-mediated Cdh1-sfGFP degradation in the skin (enveloping and epidermal basal layer). (**G**) Maximum-projected confocal images of the skin in heat-shocked *cdh1:cdh1-sfGFP; cdh1:cdh1-TagRFP* embryos transgenic for *hsp70l:zGrad* (right) or not (left) at indicated time in min after the end of heat shock (31 hpf). Scale bar: 10 μm. (**H**) Top, quantification of Cdh1-sfGFP-to-Cdh1-TagRFP ratios in the primordia of control (blue) and zGrad-expressing embryos (red) after the end of heat shock in min. Mean and SD are indicated. Bottom, Cdh1-sfGFP-to-Cdh1-TagRFP ratio of zGrad-expressing embryos normalized to control embryos (black dots) and fit to a one-exponential decay model (red). (**I**) Schematic of strategy to assess zGrad-mediated Ctnna-Citrine degradation in the skin. (**J**) Maximum-projected confocal images of skin cells in heat-shocked *ctnna:ctnna-Citrine/+* embryos transgenic for *hsp70l:zGrad* (right) or non-*hsp70l:zGrad* transgenic controls (left) at indicated time in min past the end of heat shock (31 hpf). Scale bar: 50 μm. (**K**) Left, quantification of Ctnna-Citrine levels in the skin of control (blue) and zGrad-expressing embryos (red) after the end of heat shock in min. Mean and SD are indicated. Right, Ctnna-Citrine levels in zGrad-expressing embryos normalized to Ctnna-Citrine levels in control embryos (black dots) and fit to a one-exponential decay model (red).

DOI: https://doi.org/10.7554/eLife.43125.006

**Figure 2—video 1.** Degradation of H2A-EGFP by zGrad expressed from *hsp70l* promoter.

DOI: https://doi.org/10.7554/eLife.43125.007

**Figure 2—video 2.** Degradation of Cdh1-sfGFP by zGrad expressed from *hsp70l* promoter.

DOI: https://doi.org/10.7554/eLife.43125.008

**Figure 2—video 3.** Degradation of Ctnna-Citrine by zGrad expressed from *hsp70l* promoter.

DOI: https://doi.org/10.7554/eLife.43125.009

**Figure supplement 1.** *cxcr4b* promoter activity during somitogenesis and primordium migration. (**A, B**) Images of *cxcr4b:h2a-EGFP* embryos fixed at 13 hpf (**A**) and 33 hpf (**B**) and stained by in situ hybridization against *EGFP* mRNA. (**C, D**) Images of *cxcr4b:h2a-mCherry* embryos fixed at 13 hpf (**C**) and 33 hpf (**D**) and stained by in situ hybridization against *mCherry* mRNA. Scale Bars: 200 μm.

DOI: https://doi.org/10.7554/eLife.43125.010
DOI: https://doi.org/10.7554/eLife.43125.010

**Figure supplement 2.** Fluorescent intensity of H2A-EGFP and H2A-mCherry in the somites and the primordium.

DOI: https://doi.org/10.7554/eLife.43125.011

**Figure supplement 3.** zGrad does not degrade secreted proteins.

DOI: https://doi.org/10.7554/eLife.43125.012

be a measure of H2A-EGFP in the absence of zGrad-mediated degradation. Further, we normalized the H2A-EGFP-to-H2A-mCherry fluorescence intensity ratios between zGrad-expressing and heat-shocked control embryos. In the somites, the levels of H2A-EGFP was decreased by 87% (*Figure 2C*), while in the primordium, the levels of H2A-EGFP was decreased by 22% (*Figure 2E*). The more efficient degradation of H2A-EGFP in the somites than in the primordium is probably due to the lower levels of H2A-EGFP and the lack of H2A-EGFP production in the somites at the time of zGrad induction (*Figure 2—figure supplements 1* and *2*) (*Chong et al., 2001*).

We further characterized the kinetics of zGrad-mediated degradation by determining the time interval between the start of the heat shock to induce zGrad expression and the first observable difference in EGFP/mCherry levels between zGrad-expressing embryos and control embryos. We called this time interval the time for onset of degradation. We also fitted the EGFP/mCherry ratio to a one-phase exponential decay model to extract the half-life of zGrad-mediated degradation. Although this is a simplification because the model does not account for EGFP production – a variable that we cannot easily measure – we expect that it gives a rough estimate for the time it takes to degrade a GFP-tagged protein once zGrad is expressed. The time for onset of degradation and half-life for H2A-EGFP in the somites was 200 min and 156 min, respectively (*Figure 2C*, *Table 1*). The time for onset of degradation and the half-life for H2A-EGFP degradation in the primordium was 140 min and 29 min, respectively (*Figure 2E*, *Table 1*).

Second, we tested whether zGrad degrades the transmembrane protein E-Cadherin (Cdh1). We generated two transgenes that express Cdh1-sfGFP and Cdh1-TagRFP from a 72 kb genomic fragment spanning the *cdh1* locus (*cdh1:cdh1-sfGFP; cdh1:cdh1-TagRFP*). These transgenes recapitulated the endogenous Cdh1 expression pattern and rescued the lethality of *cdh1* mutant embryos (*Table 2*), indicating that Cdh1-sfGFP and Cdh1-TagRFP are functional. Using these lines, we generated *hsp70l:zGrad; cdh1:cdh1-sfGFP; cdh1:cdh1-TagRFP* embryos and *cdh1:cdh1-sfGFP; cdh1:cdh1-*

**Table 1.** *Cdh1* transgenic lines rescue *cdh1* mutants.

| Transgenic line | Total number of embryos | Transgenic embryos phenotypically wild type | Non-transgenic embryos phenotypically wild type | Embryos phenotypically *cdh1* mutant |
|---|---|---|---|---|
| *cdh1:cdh1-sfGFP* | 306 (100 %) | 149 (47 %) | 96 (33 %) | 61 (20 %) |
| *cdh1:cdh1-TagRFP* | 432 (100 %) | 204 (47 %) | 172 (40 %) | 56 (13%) |

DOI: https://doi.org/10.7554/eLife.43125.013

*TagRFP* control embryos. These embryos were heat shocked at 31 hpf for 30 min and imaged for 4.8 hr. In zGrad-expressing embryos, Cdh1-sfGFP degradation was uniformly detected within 15 min post heat shock in the skin of the embryo (enveloping layer and epidermal basal layer) (*Figure 2F–H*, *Figure 2—Video 2*). As detailed above for the quantification of zGrad-mediated H2A-EGFP degradation, we used the fluorescence intensity of Cdh1-TagRFP as a reference to quantify the reduction of Cdh1-sfGFP. We divided the Cdh1-sfGFP fluorescence intensity by the Cdh1-TagRFP fluorescence intensity and normalized the ratio in the zGrad-expressing embryos to the ratio in the heat-shocked control embryos. This analysis showed that zGrad expression reduced the levels of Cdh1-sfGFP in the primordium by 79% with a degradation half-life of 24 min (*Figure 2H*). The time for onset of Cdh1-sfGFP degradation was 75 min (*Table 1*).

Third, we tested whether zGrad degrades the cytoplasmic protein αE-Catenin (Ctnna). To address this, we used a gene-trap line that expresses αE-Catenin-Citrine (Ctnna-Citrine) from the *ctnna* locus (*Trinh et al., 2011*; *Žigman et al., 2011*). In this line an artificial exon containing the *Citrine* coding sequence flanked by a splice donor and a splice acceptor is inserted between the exons 7 and 8 of the *ctnna* gene (*Žigman et al., 2011*). Homozygous *ctnna:ctnna-Citrine* fish are viable, indicating that Ctnna-Citrine is functional (*Trinh et al., 2011*). We generated *hsp70l:zGrad; ctnna:ctnna-Citrine/+* embryos and *ctnna:ctnna-Citrine/+* control embryos and heat shocked these embryos at 31 hpf for 30 min and imaged the embryos for 8 hr (*Figure 2I*). In contrast to heat-shocked control embryos, Ctnna-Citrine was degraded in zGrad-expressing embryos (*Figure 2J,K*). Since Ctnna tagged with a red fluorescent protein is not available, we could not perform a ratiometric analysis and instead quantified the total Ctnna-Citrine fluorescent intensity. This analysis showed that the time for onset of degradation of Ctnna-Citrine was 65 min and that Ctnna-Citrine was degraded by 58% with a half-life of 19 min in zGrad-expressing embryos (*Figure 2K*, *Table 1*).

Finally, we tested whether zGrad degrades GFP targeted to the secretory pathway by a signal peptide. Using fish that express secreted EGFP and secreted mCherry from a heat shock promoter (*hsp70l:sec-GFP; hsp70:sec-mCherry*), we generated *hsp70l:sec-GFP; hsp70:sec-mCherry; hsp70l:zGrad* embryos and *hsp70l:sec-GFP; hsp70:sec-mCherry* control embryos. Unlike H2A-EGFP, Cdh1-sfGFP and Ctnna-Citrine, expression of zGrad did not degrade secreted GFP (*Figure 2—figure supplement 3*).

Together, these observations indicate that zGrad targets nuclear, cytoplasmic, and transmembrane proteins tagged with EGFP or Citrine for degradation.

**Table 2.** Summary of degradation kinetics.

| FP-Tagged protein | Promoter expressing zGrad | Percent reduction | Degradation half-life in min | Time for onset of degradation in min |
|---|---|---|---|---|
| sfGFP-ZF1 | mRNA | 89% | N/A | N/A |
| Cxcr4b-EGFP | *cxcr4b* | 86% | N/A | N/A |
| H2A-EGFP (in somites) | *hsp70l* | 87% | 156 | 200 |
| H2A-EGFP (in primordium) | *hsp70l* | 22% | 29[†] | 140 |
| Cdh1-sfGFP | *hsp70l* | 79% | 24 | 75 |
| Ctnna-Citrine | *hsp70l* | 58% | 19 | 65 |

[†]Note that the zGrad-mediated degradation of H2A-EGFP in the primordium is obscured by the continued production of H2A-EGFP.

DOI: https://doi.org/10.7554/eLife.43125.014

## zGrad-mediated protein degradation results in loss-of-protein function

Since zGrad efficiently degrades proteins from different cellular compartments (*Figure 2*), we asked whether we can use zGrad-mediated protein depletion to replicate known loss of protein function phenotypes and uncover novel ones. We addressed this question in several ways. First, we tested whether zGrad degrades cytoplasmic Ctnna-Citrine to disrupt its function in cell-cell adhesion (*Kofron et al., 1997*) using the *ctnna:ctnna-Citrine* gene trap line. We injected *zGrad* mRNA or *sfGFP* control mRNA into one-cell stage embryos that were maternal-zygotic Ctnna-Citrine (*MZ ctnna:ctnna-Citrine*, *Figure 3A*), maternal Ctnna-Citrine (*M ctnna:ctnna-Citrine*, *Figure 3D*) or zygotic Ctnna-Citrine (*Z ctnna:ctnna-Citrine*, *Figure 3G*). In *zGrad* mRNA-injected *MZ ctnna:ctnna-*

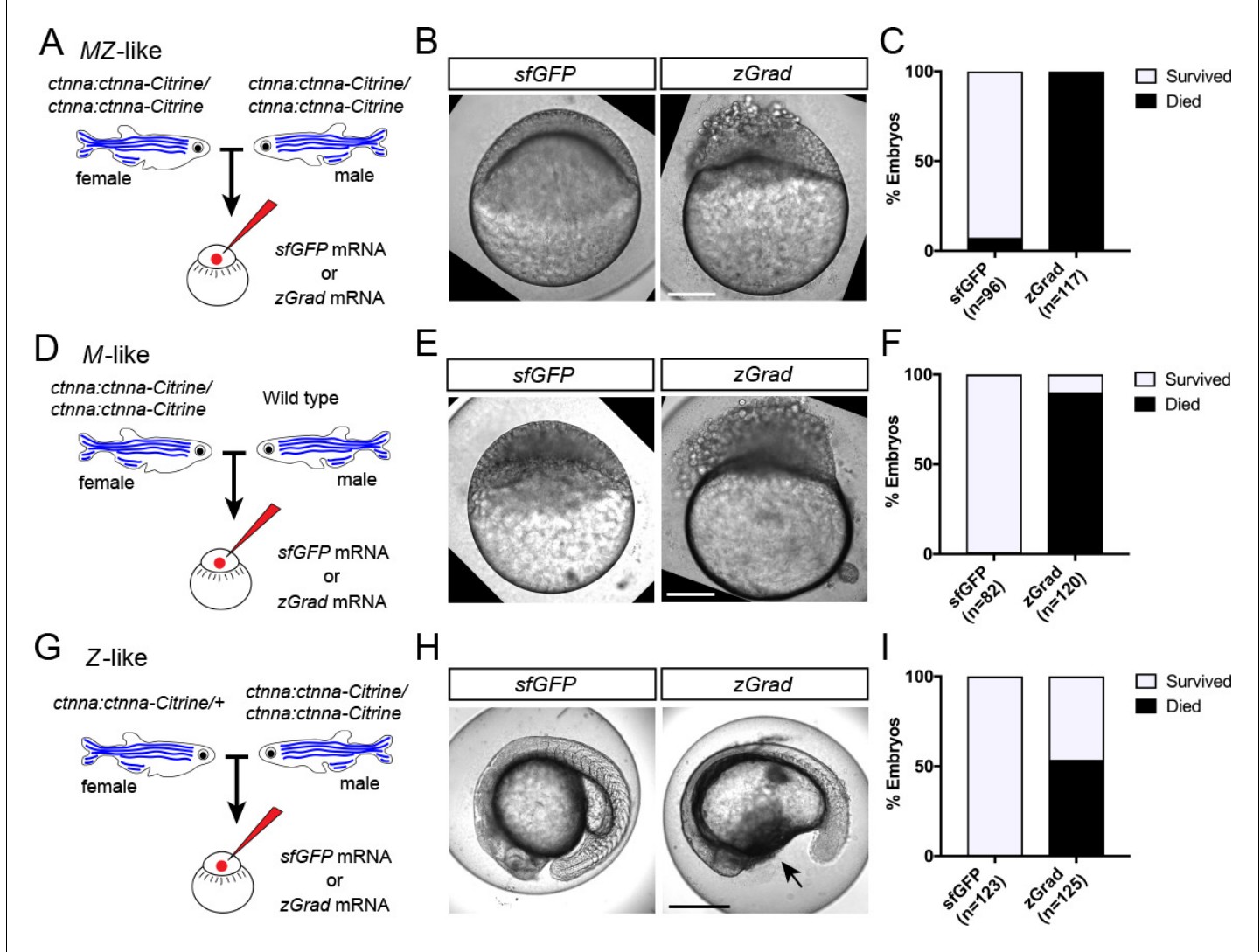

**Figure 3.** zGrad-mediated depletion of alpha-Catenin results in cell adhesion defects. (**A**) Breeding strategy to assess zGrad-mediated degradation of maternally and zygotically provided Ctnna-Citrine on embryonic development. (**B**) Images of MZ *ctnna:ctnna-Citrine* embryos injected with *sfGFP* control mRNA (left) or *zGrad* mRNA (right). Scale bar: 100 μm. (**C**) Quantification of MZ *ctnna:ctnna-Citrine* embryos injected with *sfGFP* control mRNA or *zGrad* mRNA that disintegrated and died. (**D**) Breeding strategy to assess zGrad-mediated degradation of maternally provided Ctnna-Citrine on embryonic development. (**E**) Images of M *ctnna:ctnna-Citrine* embryos injected with *sfGFP* control mRNA (left) or *zGrad* mRNA (right). Scale bar: 100 μm. (**F**) Quantification of M *ctnna:ctnna-Citrine* embryos injected with *sfGFP* control mRNA or *zGrad* mRNA that disintegrated and died. (**G**) Breeding strategy to assess zGrad-mediated degradation of zygotically provided Ctnna-Citrine on embryonic development. (**H**) Images of Z *ctnna:ctnna-Citrine* embryos injected with *sfGFP* control mRNA (left) or *zGrad* mRNA (right). Scale bar: 100 μm. (**I**) Quantification of Z *ctnna:ctnna-Citrine* embryos injected with *sfGFP* control mRNA and *zGrad* mRNA that displayed tissue rupture (arrow) and died.
DOI: https://doi.org/10.7554/eLife.43125.015

*Citrine* and *M ctnna:ctnna-Citrine* embryos the cells started to become detached and partly shed from the embryo resulting in embryonic lethality around 3 to 5 hpf (*Figure 3B,C,E,F*). Control embryos injected with *sfGFP* mRNA developed normally (*Figure 3B,C,E,F*). Similarly but delayed by 20 hr, in *zGrad* mRNA-injected *Z ctnna:ctnna-Citrine* embryos, cells detached from the embryo by 24 hpf and about half of the embryos died (note, half the embryos were zygotically homozygous for *ctnna:ctnna-Citrine* since we crossed *ctnna:ctnna-Citrine/ctnna:ctnna-Citrine* males to *ctnna:ctnna-Citrine/+* females, *Figure 3G,H,I*) while control injected embryos were unaffected (*Figure 3H,I*). The observed phenotype in *Z ctnna:ctnna-Citrine* embryos injected with *zGrad* mRNA is similar to the defects reported for zygotic *ctnna-/-* embryos (*Han et al., 2016*). Thus, zGrad depletes Ctnna-Citrine to levels too low to sustain cell-cell adhesion and embryonic development. Moreover, it suggests that maternally supplied Ctnna can sustain cell-cell adhesion and embryonic development for about 1 day before the embryo requires zygotic Ctnna.

Second, we tested whether zGrad degrades transmembrane Cdh1-sfGFP efficiently enough to disrupt its function (*Yoshida-Noro et al., 1984*; *Stephenson et al., 2010*; *Larue et al., 1994*) and uncover late requirements of Cdh1 in cell-cell adhesion. Such an analysis is currently not possible because *cdh1-/-* embryos die during late gastrulation stages (*Kane et al., 1996*; *Kane et al., 2005*; *Montero et al., 2005*). To address this, we crossed *hsp70l:zGrad/+; cdh1+/-; cdh1:cdh1-sfGFP/+* fish to *cdh1+/-* fish and as a control *cdh1+/-; cdh1:cdh1-sfGFP/+* fish to *cdh1+/-* fish. We separated *cdh1:cdh1-sfGFP* transgenic from *cdh1:cdh1-sfGFP* non-transgenic embryos from both crosses and separately heat-shocked the embryos for 1 hr at 25 hpf (*Figure 4A*, *Figure 4—figure supplement 1A*). In contrast to control embryos which showed no visible defects (*Figure 4B,C*), we found *cdh1-/-; cdh1:cdh1-sfGFP/+; hsp70l:zGrad/+* embryos shed their skin and rapidly died (*Figure 4D* and *Figure 4—figure supplement 1A*), consistent with the idea that zGrad-mediated Cdh1-sfGFP degradation disrupts the cell-cell adhesion between the skin cells of the embryos that lack endogenous Cdh1 function. We would like to note that the skin phenotype in embryos with depletion of Cdh1-sfGFP was not always fully penetrant (*Figure 4—figure supplement 1A*). Depending on the stage and the duration of the heat shock, we believe that Cdh1-sfGFP is not always depleted to low enough levels to cause a cell-cell adhesion defect. In summary, these observations suggest that transient expression of zGrad can efficiently degrade GFP-tagged proteins and, if GFP-tagged proteins are depleted sufficiently, cause loss-of-protein function and uncover phenotypes past the initial requirements of essential proteins.

Third, we asked whether induction of zGrad can degrade the Sdf1 chemokine receptor Cxcr4b tagged with EGFP to temporarily disrupt its function as a guidance receptor for the migration of the primordium (*Haas and Gilmour, 2006*; *David et al., 2002*). For this, we heat shocked *hsp70l:zGrad; cxcr4b-/-; cxcr4b:cxcr4b-EGFP-IRES-Kate2-CaaX-p7* embryos and *cxcr4b-/-; cxcr4b:cxcr4b-EGFP-IRES-Kate2-CaaX-p7* control embryos at 28 hpf when the primordium has completed a third of its migration along the body of the embryo and recorded the levels of Cxcr4b-EGFP and the migration of the primordium by time lapse microscopy. Cxcr4b-EGFP expressed from the *cxcr4b:cxcr4b-EGFP-IRES-Kate2-CaaX-p7* transgene restores the migration of the primordium in *cxcr4b-/-* embryos and Kate2-CaaX expressed from an internal ribosomal entry site (IRES) serves as a marker for the primordium and an internal reference for how much Cxcr4b-EGFP is produced (*Venkiteswaran et al., 2013*; *Lewellis et al., 2013*). This analysis showed that induction of zGrad degraded Cxcr4b-EGFP (*Figure 5A,B,E*) and resulted in rounded primordium morphology (*Figure 5A,C*) and stalled primordium migration (*Figure 5A,D,E*) 90 min after heat shock. After 120 min newly produced Cxcr4b-EGFP was detectable again and the primordium resumed its migration (*Figure 5A,E*, *Figure 5— Video 1*). Thus, a pulse of zGrad expression can deplete Cxcr4b-EGFP to levels unable to sustain directed cell migration of the primordium. This suggests that temporal zGrad induction from the heat shock promoter can be used to induce reversible protein loss-of-function scenarios.

Fourth, we asked whether tissue-specific expression of zGrad from the *cxcr4b* promoter can degrade Cxcr4b-EGFP and recapitulate the *cxcr4b* mutant primordium migration phenotype (*Haas and Gilmour, 2006*). To address this question, we expressed zGrad from a 69 kb genomic fragment spanning the *cxcr4b* locus (*cxcr4b:zGrad*) (*Figure 6A*). The expression pattern of *zGrad* mRNA in *cxcr4b:zGrad* embryos faithfully recapitulated the *cxcr4b* mRNA expression pattern (*Figure 6B*). We crossed this line into the *cxcr4b:cxcr4b-EGFP-IRES-Kate2-CaaX-p1; cxcr4b-/-* background and determined the degree of Cxcr4b-EGFP degradation compared to Kate2-CaaX expression in primordia of *cxcr4b:zGrad* embryos and control embryos. Similar to the *cxcr4b:cxcr4b-EGFP-*

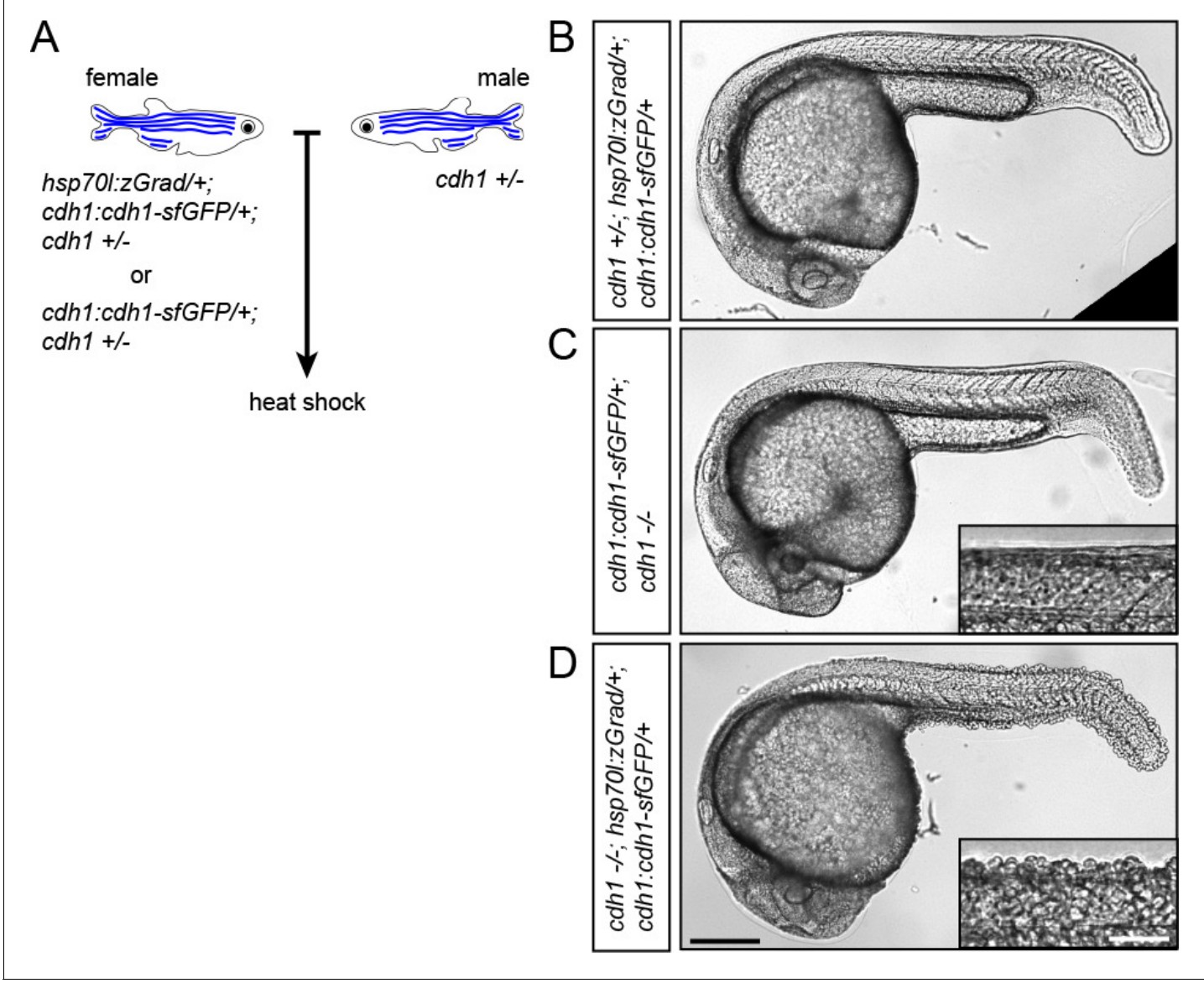

**Figure 4.** zGrad-mediated depletion of Cadherin-1 at 25 hpf results in skin defects and lethality. (A) Breeding strategy to assess heat shock-induced zGrad-mediated degradation of Cdh1-sfGFP on embryonic development. (B, C, D) Images of *cdh1+/-; hsp70l:zgrad/+; cdh1:cdh1-sfGFP/+* control embryo (B), *cdh1-/-; cdh1:cdh1-sfGFP/+* control embryo (C) and *cdh1-/-; hsp70l:zGrad/+; cdh1:cdh1-sfGFP* embryo with skin defects (D) at 29 hpf. Embryos were heat shocked at 25 hpf for 1 hr. Scale bar: 200 μm. Insets in (C, D) are magnified images of the skin in the same embryos. Scale bar: 50 μm.

DOI: https://doi.org/10.7554/eLife.43125.016

The following figure supplement is available for figure 4:

**Figure supplement 1.** Expected results for zGrad-induced depletion of sfGFP-tagged Cadherin-1.
DOI: https://doi.org/10.7554/eLife.43125.017

*IRES-Kate2-CaaX-p7* transgenic line used above, the *cxcr4b:cxcr4b-EGFP-IRES-Kate2-CaaX-p1* fully restores primordium migration in *cxcr4b* mutant embryos, however, it expresses Cxcr4b-EGFP at two-fold lower levels (*Fuentes et al., 2016*). This analysis showed that zGrad efficiently depleted Cxcr4b-EGFP below detectable levels (*Figure 6C,D*). Next, we assessed the primordium migration by staining for *cxcr4b* mRNA at 38 hpf. In *cxcr4b* mutant control embryos that carried the rescuing *cxcr4b:cxcr4b-EGFP-IRES-Kate2-CaaX-p1* transgene but lacked the *cxcr4b:zGrad* transgene, the primordium completed its migration and deposited the same number of neuromasts with the same

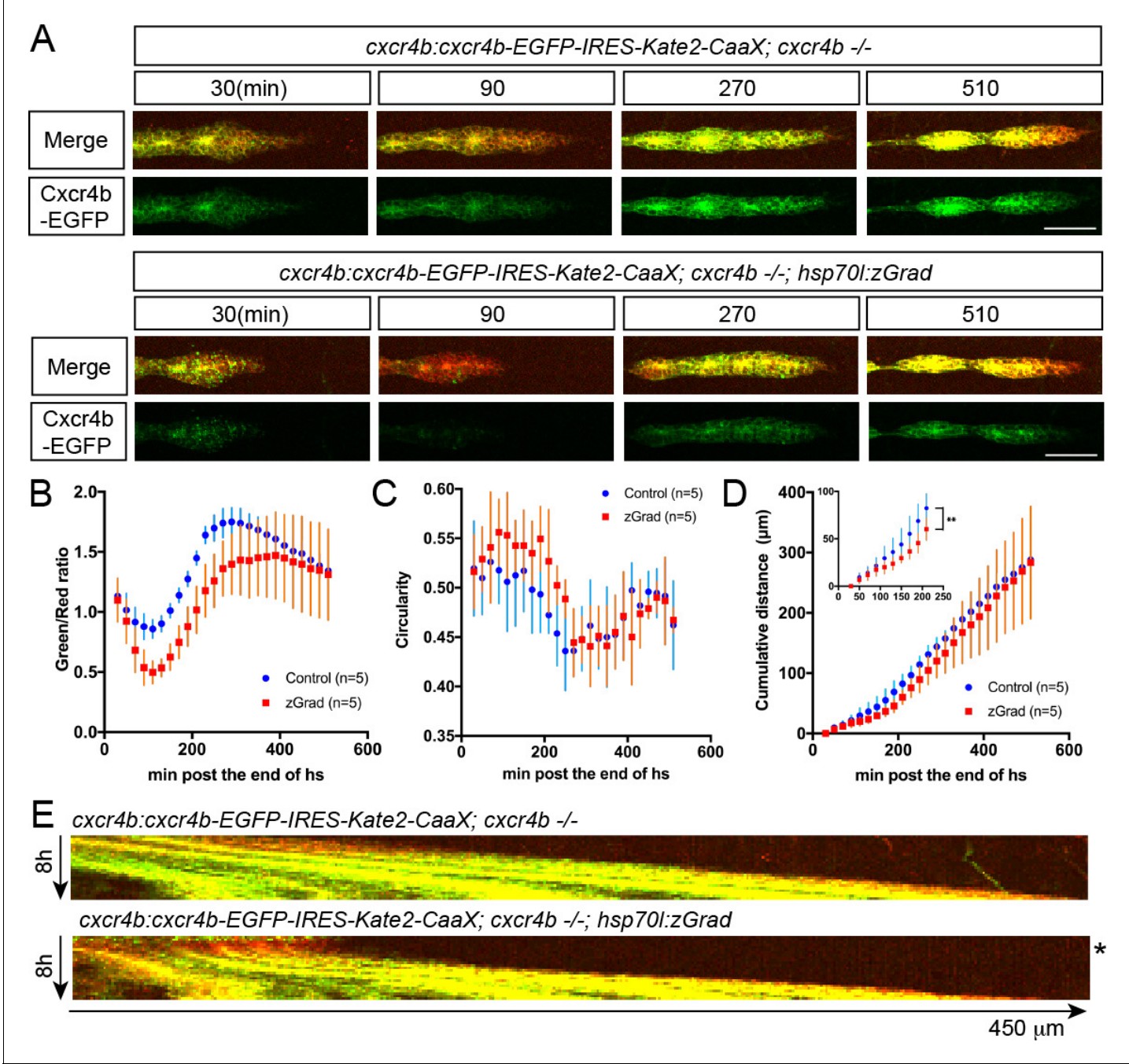

**Figure 5.** A pulse of zGrad degrades Cxcr4b-GFP and stalls primordium migration transiently. (A) Maximum-projected confocal images of the primordia in heat-shocked *cxcr4b:cxcr4b-EGFP-IRES-Kate2-CaaX-p7; cxcr4b-/-* embryos transgenic for *hsp70l:zGrad* (lower panels) or not (upper panels) at indicated time in min after the end of heat shock (31 hpf). Scale bar: 50 μm. (B) Quantification of Cxcr4b-EGFP-to-Kate2-CaaX ratios in the primordia of control (blue) and zGrad-expressing embryos (red) after the end of heat shock in min. Mean and SD are indicated. (C) Quantification of circularity of the primordia of control (blue) and zGrad-expressing embryos (red) after the end of heat shock in min. Mean and SD are indicated. (D) Quantification of the cumulative primordium migration distance in control (blue) and zGrad-expressing embryos (red) after the end of heat shock in min. Mean and SD are indicated. Inset shows magnification of the 30 min to 240 min time interval. **=p < 0.01. (E) Kymograph of the primordia in heat-shocked *cxcr4b: cxcr4b-EGFP-IRES-Kate2-CaaX-p7; cxcr4b-/-* embryos transgenic for *hsp70l:zGrad* (bottom) or not (top). Cxcr4b-EGFP is shown in green and Kate2-CaaX in red. Asterisk indicates the time interval in which Cxcr4b-EGFP is transiently degraded and the primordium transiently ceases to migrate.
DOI: https://doi.org/10.7554/eLife.43125.018

The following video is available for figure 5:

**Figure 5—video 1.** Degradation of Cxcr4b-EGFP by zGrad expressed from *hsp70l* promoter.

*Figure 5 continued on next page*

Figure 5 continued

DOI: https://doi.org/10.7554/eLife.43125.019

spacing as in wild-type embryos (*Figure 6E,F*, *Figure 6—figure supplement 1B*). In contrast, embryos with the *cxcr4b:zGrad* transgene migrated on average only 72.5% of the distance (*Figure 6E,F*). Compared to primordia in *cxcr4b-/-* embryos, which showed little to no directed migration (*Figure 6E,F*) (*Haas and Gilmour, 2006*), primordia in *cxcr4b-/-; cxcr4b:cxcr4b-EGFP-IRES-Kate2-CaaX-p1; cxcr4b:zGrad* embryos migrated directionally but at reduced speed, did not fully complete their migration and failed to deposit the terminal neuromasts by 4 dpf (*Figure 6E,F*, *Figure 6—figure supplement 1B,C*). Importantly, expression of zGrad in the primordium did not affect its migration (*Figure 6—figure supplement 1A*). This suggests that zGrad expression from the *cxcr4b* promoter efficiently degrades Cxcr4b-EGFP to levels low enough to slow primordium migration but not low enough to recapitulate the primordium migration defect observed in embryos with loss of Cxcr4b function, probably because the *cxcr4b* promoter does not express zGrad at high enough levels for complete degradation of Cxcr4b-EGFP driven from the same promoter. More generally, these observations indicate that expression of zGrad from a tissue-specific promoter can result in efficient degradation of GFP-tagged proteins to levels low enough to perturb protein function and cause tissue-specific defects.

## Discussion

In our study, we sought to develop a tool that allows for the acute inactivation of proteins in zebrafish. We modified the anti-GFP nanobody-based deGradFP system from flies (*Brauchle et al., 2014*) and adapted it to zebrafish and named it zGrad. zGrad efficiently degrades GFP-tagged transmembrane, cytoplasmic and nuclear proteins. It recognizes different GFP versions (EGFP, sfGFP and Citrine, *Figures 1* and *2*) and targets the tagged proteins for degradation. In contrast to other degron systems (*Natsume and Kanemaki, 2017*) and similar to deGradFP (*Caussinus et al., 2011*), zGrad degraded proteins tagged at the N-terminus, embedded within the protein and at the C-terminus in the examples reported here (*Figures 1* and *2*). We found that zGrad rapidly degrades GFP-tagged proteins with half-lives of around 20 min for transmembrane and cytoplasmic proteins (*Figure 2* and *Table 1*). These kinetics are similar to the kinetics reported for other degron-based systems (*Daniel et al., 2018*; *Nishimura et al., 2009*; *Armenti et al., 2014*). The degradation of H2A-EGFP was significantly slower displaying a half-life of about 2.5 hr (*Figure 2* and *Table 1*). This could be due to the long life-time of histone proteins (*Clift et al., 2017*) and the possibility that protein degradation is less efficient in nuclei. Importantly, in our system zGrad expression needs to be induced. This delays the onset of degradation by 65 to 200 min for the proteins we investigated. Nevertheless, zGrad degrades proteins rapidly enough to assess the consequences of abrupt protein inactivation.

Recently, the AID system was combined with the deGradFP system by fusing AID to the vhhGFP4 anti-GFP nanobody and shown to degrade GFP-tagged proteins in an auxin-dependent manner in zebrafish (*Daniel et al., 2018*). This elegant approach adds temporal control to the spatial control provided by the deGradFP system. Also, the fusion of AID to the vhhGFP4 nanobody overcomes the leakiness that we observed with the AID system (*Daniel et al., 2018*). However, compared to zGrad system, which requires the fusion of the protein of interest to GFP and the expression of zGrad, the mAID-nanobody system requires the expression of an additional component, the mAID-nanobody fusion protein – a feature that adds more versatility but also requires additional effort.

One simple use of zGrad is the study of the contribution of maternally supplied proteins in early development. Depletion of maternal proteins by expressing zGrad from injected mRNA or as purified protein should uncover early protein requirements and can circumvent the laborious process of generating maternal mutants for essential genes by germline replacement (*Ciruna et al., 2002*). Our finding that the depletion of maternal Ctnna results in an early and severe cell-cell adhesion defect is consistent with this idea (*Figure 3*). Alternatively, a pulse of zGrad should allow one to dissect the temporal requirements of proteins. Depending on the protein function this could result in a reversible phenotype as observed when we transiently stalled the migration of the primordium by depleting Cxcr4b (*Figure 5*) or to an irreversible phenotype if the function of the protein is continually

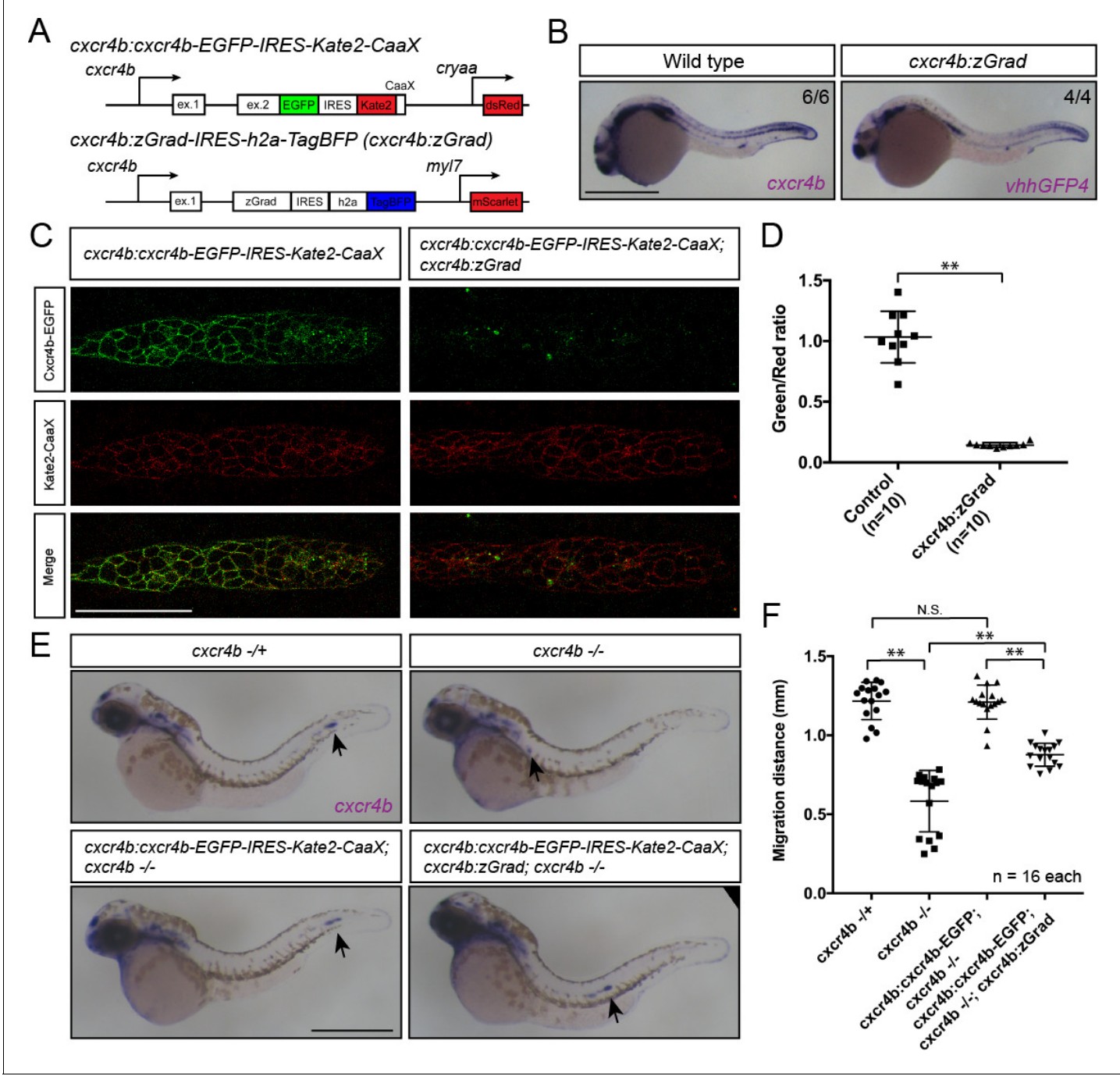

**Figure 6.** Tissue-specific expression of zGrad in the primordium degrades Cxcr4b-EGFP and slows down primordium migration. (A) Schematic of strategy to assess zGrad-mediated Cxcr4b-EGFP degradation in the primordium on primordium migration. (B) In situ hybridization against *cxcr4b* mRNA in a wildtype embryo and against *zGrad* mRNA in a *cxcr4b:zGrad* embryo at 24hpf. Scale bar: 0.5 mm. (C) Single-plane confocal images of the primordium in *cxcr4b:cxcr4b-EGFP-IRES-Kate2-CaaX-p1* control (left) and *cxcr4b:cxcr4b-EGFP-IRES-Kate2-CaaX-p1; cxcr4b:zGrad* embryos (right) at 36 hpf. Note that the embryos are *cxcr4b +/-* or *cxcr4b -/-*. Scale bar: 50 µm. (D) Quantification of Cxcr4b-EGFP to Kate2-CaaX fluorescence intensity ratio in the primordia of control embryos (blue) and embryos expressing zGrad in the primordium at 36 hpf. Mean and SD are indicated. **=p < 0.01. (E) In situ hybridization against *cxcr4b* mRNA in *cxcr4b-/+* (top left), *cxcr4b-/-* (top right), *cxcr4b:cxcr4b-EGFP-IRES-Kate2-CaaX-p1; cxcr4b-/-* (bottom left) and *cxcr4b:cxcr4b-EGFP-IRES-Kate2-CaaX-p1; cxcr4b-/-; cxcr4b:zGrad* embryos (bottom right) at 38 hpf. Arrows indicate the location of the primordium. Scale bar: 0.5 mm. (F) Quantification of primordia migration distance of the indicated genotypes at 38 hpf. Mean, SD and n are indicated. **=p < 0.01, N.S. = p > 0.05.

DOI: https://doi.org/10.7554/eLife.43125.020

*Figure 6 continued on next page*

*Figure 6 continued*

The following figure supplement is available for figure 6:

**Figure supplement 1.** Tissue-specific Cxcr4b-EGFP degradation by zGrad.

DOI: https://doi.org/10.7554/eLife.43125.021

required for viability as observed when we depleted Cdh1 and disrupted the integrity of the embryo (*Figure 4*). Another use of zGrad is to deplete proteins from certain tissues. By expressing zGrad from tissue-specific promoters, one can circumvent essential early requirements, assess later protein functions and disentangle the contribution of tissues to complex phenotypes. Our observation that depletion of Cxcr4b in the primordium by expressing zGrad from the *cxcr4b* promoter results in slowed migration supports this idea (*Figure 6*).

An important consideration is that zGrad needs to be expressed at high enough levels to deplete proteins to sufficiently low levels to fully disrupt protein function. One way to achieve high and tissue-specific zGrad expression is to amplify the production of zGrad through such systems as Gal4/UAS (*Scheer and Campos-Ortega, 1999*) or – to also add temporal control – inducible systems such as the Tet-On system (*Campbell et al., 2012*). Such modifications combined with the increasing number of GFP-tagged transgenic (*Trinh et al., 2011*; *Kawakami et al., 2010*) and knock-in lines (*Hoshijima et al., 2016*) should render zGrad as a useful tool for the study of the consequences of acute loss-of-protein function in zebrafish, possibly also circumventing the problem of genetic compensation observed in studies of genetic mutants (*Rossi et al., 2015*).

## Materials and methods

### Zebrafish strains

Embryos were staged as previously described (*Kimmel et al., 1995*). Hours post fertilization (hpf) was used to determine the developmental age of the embryos. Embryos were incubated at 28.5°C from one-cell-stage (0 hpf) until the indicated time. *cxcr4b*$^{t26035}$ (*Knaut et al., 2003*) homozygous mutant embryos were generated by inbreeding heterozygous adults, crossing homozygous adult with heterozygous adults or inbreeding homozygous adults. *tg(cxcr4b:cxcr4b-EGFP-IRES-Kate2-CaaX)* (*Venkiteswaran et al., 2013*; *Lewellis et al., 2013*) embryos were generated by crossing heterozygous adults with wild-type adults and sorted by GFP expression. *Gt(ctnna-citrine)Ct3a* (*Trinh et al., 2011*; *Žigman et al., 2011*) lines were maintained by inbreeding of heterozygous adults or homozygous adults. *cdh1(tx230)* (*Kane et al., 1996*) were kept as heterozygous adults and *cdh1* homozygous embryos were generated by inbreeding heterozygous adults. The *hsp70:sec-mCherry* (*Wang et al., 2018*), *cxcr4b:cxcr4b-EGFP-IRES-Kate2-CaaX-p1* (*Lewellis et al., 2013*), *cxcr4b: cxcr4b-EGFP-IRES-Kate2-CaaX-p7* (*Venkiteswaran et al., 2013*), *cxcr4b:h2a-EGFP* (*Kozlovskaja-Gumbrienė et al., 2017*) and *cxcr4b:h2a-mCherry* (*Wang et al., 2018*) were previously described. Zebrafish were maintained under the approval from IACUC (protocol number: 170105–02).

### Generation of transgenic animals

#### hsp70l:zGrad-IRES-h2a-TagBFP

*zGrad*, *IRES-h2a* and *TagBFP* sequences were amplified by PCR and cloned into the plasmid *pDEST-tol2-hsp70l* (*Kwan et al., 2007*) by Gibson assembly (*Gibson et al., 2009*). *pDEST-tol2-hsp70l* contains a genomic fragment spanning 1.5 kb upstream of the *hsp70l* start codon (*Halloran et al., 2000*). 50 ng/µl of the *pDEST-tol2-hsp70l-zGrad-IRES-TagBFP* plasmid was co-injected with 25 ng/µl *tol2* mRNA into one-cell-stage embryos. Founder fish were identified by in situ hybridization against *vhhgfp4* sequence of heat-shocked offspring. Briefly, adult fish that were injected as embryos were outcrossed with wild-type adults and embryos were collected. At the shield stage, the embryos were heat-shocked at 39.5°C for 30 min. Preparation of the antisense RNA probe and whole-mount in situ hybridization were performed as previously described (*Knaut et al., 2003*). Fish that gave rise to *hsp70l:zGrad-IRES-H2A-TagBFP* offspring were kept as founder fish. zGrad activity was confirmed by heat shocking *hsp70l:zGrad-IRES-H2A-TagBFP; cxcr4b:h2a-EGFP* embryos and observation of H2A-GFP degradation. The line from the founder that showed the strongest GFP degradation was

kept and used in this study. The full name of this line is *tg(hsp70l:zGrad-IRES-h2a-TagBFP)p1*. Note H2A-TagBFP expression from the IRES is not detectable in this line.

## cxcr4b:zGrad-IRES-h2a-TagBFP

For the *cxcr4b:zGrad-IRES-h2a-TagBFP* BAC transgene, we used the BAC clone DKEY-169F10 (*Suster et al., 2011*). This BAC clone contains the *cxcr4b* locus. We modified this BAC in two ways by recombineering. First, we modified *pBS-IndHom-Tol2-FRT-GalK-cryaa-dsRed* (Addgene plasmid # 73203 (*Fuentes et al., 2016*)) by replacing the *cryaa:dsRed* sequence with *myl7:mScarlet* and the *FRT-GalK-FRT* sequence with the *kanR* sequence. The *myl7* (synonym *cmlc2*) promoter drives expression in the myocardium (*Huang et al., 2003*) and *kanR* allows for rapid kanamycin-based selection of recombinants. The full name of this plasmid is *pBS-IndHom-Tol2(exon4)-Kan-Tol2(exon1)- myl7: mScarlet-Hom*. We amplified *Tol2(exon4)-Kan-Tol2(exon1)- myl7:mScarlet* cassette by PCR and inserted it into the *pIndigoBAC-5* backbone of the DKEY-169F10 BAC clone using *kanamycin* as a selection marker. Second, a cassette containing *zGrad-IRES-h2a-TagBFP-FRT-galK-FRT* flanked by homology arms upstream of the *cxcr4b* exon2 and downstream of the *cxcr4b* stop codon was inserted to replace the *cxcr4b* coding sequence in exon2 using *galK* as a selection marker. The *galK* cassette was removed by Flippase. The obtained BAC was characterized with sequencing the modified region in the BAC and EcoRI finger printing. This transgene expresses zGrad-IRES-H2A-TagBFP fused to five amino acids from *cxcr4b* exon1 under the *cxcr4b* promoter. The BAC was prepared with the nucleobond BAC 100 kit (Clonetech) and co-injected with 25 ng/µl *tol2* mRNA into one-cell-stage embryos. Stable transgenic animals were identified by out-crossing injected adults and screening for the red hearts in 4 dpf larvae. Embryos were collected from each stable transgenic fish and tested for expression of *zGrad* by whole mount in situ hybridization 30 hpf. We selected the transgenic line that expressed the highest level of *zGrad* from the BAC transgene and used it in this study. The full name of this transgenic line is *TgBAC(cxcr4b:zGrad-IRES-h2a-TagBFP)p3*. Note H2A-TagBFP expression from the IRES is not detectable in this line.

For the *hsp70l:sec-GFP* transgene, the *pDestTol2pA-hsp70l-anos1b-sv40pA* plasmid was used as a template (*Wang et al., 2018*). Using Gibson cloning (*Gibson et al., 2009*), we replaced the *anos1b* coding sequence with the coding sequence of *GFP* fused to the 3' end of the *fgf3* secretion signal (amino acids 1 to 18). We verified the final construct by sequencing and co-injected it along with *tol2* transposase mRNA into zebrafish embryos at the one-cell stage. Stable transgenic fish were identified by out-crossing adults injected with the transgene and raising larvae from fish whose offspring were identified to express GFP upon heat shock as determined by green fluorescence. Founder fish were verified to carry a single copy of the transgene by determining the fraction of progeny carrying the transgene. The full name of this transgenic line is *Tg(hsp70l:sec-GFP)p1*.

## cdh1:cdh1-sfGFP and cdh1:cdh1-TagRFP lines

For the *cdh1:cdh1-sfGFP* and *cdh1:cdh1-TagRFP* BAC transgenes, we used the BAC clone CHORI-211–175 C23. This BAC clone spans 72 kb of genomic sequence and contains the *cdh1* locus and was modified in two ways by recombineering. First, the *Tol2* sites and the *cryaa:Cerulean* transgenesis marker were inserted into the BAC backbone (*Fuentes et al., 2016*). Second, a cassette consisting of *sfGFP-FRT-galK-FRT* or *TagRFP-FRT-galK-FRT* flanked by 446 bp and 590 bp of homology upstream of the stop codon in *cdh1* exon 16, and downstream of the stop codon in *cdh1* exon 16, respectively, was inserted before the *cdh1* stop codon using galK-mediated recombineering (*Warming et al., 2005*). The galK cassette was removed by Flippase-mediated recombination. This transgene expresses the full length of Cdh1 fused to sfGFP or TagRFP from the *cdh1* promoter. The final BAC transgenes were characterized by EcoRI restriction digestion and sequencing of PCR amplicons of the modified locus. The CHORI-211–175 C23 BAC clone was obtained from BACPAC Resources, Children' Hospital Oakland Research Institute, CA (bacpacorders@chori.org). The BACs were purified with the nucleobond BAC 100 kit (Clontech). We co-injected 1 nl of 40 ng/µl *Tol2* mRNA and 50–250 ng/µl of the *cdh1:cdh1-sfGFP* or *cdh1:cdh1-TagRFP* BAC transgene DNA into the lifting cell of the zygote of 0 to 20 min old embryos. The *Tol2* mRNA was transcribed from the pCS2FA-transposase plasmid (*Kwan et al., 2007*) using the mMessage mMachine SP6 Transcription Kit (Thermo Fisher). Stable transgenic larvae were identified by out-crossing adults injected with the *cdh1:cdh1-sfGFP* or *cdh1:cdh1-TagRFP* BAC transgenes, and by raising larvae positive for the blue

fluorescent transgenesis marker in the lens of the eye at 4 dpf. The full names of these two transgenic lines are *TgBAC(cdh1:cdh1-sfGFP)p1* and *TgBAC(cdh1:cdh1-TagRFP)p1*.

All transgenic lines generated for this study are available upon request.

## Generation of plasmids for in vitro transcription

To construct *pCS2+-mScarlet-V5*, the *mScarlet* coding sequence was amplified from *pmScarlet-C1* (Addgene plasmid # 85042, (*Bindels et al., 2017*)) by PCR with a 5' primer containing V5-tag sequence and cloned into *pCS2+* plasmid by Gibson assembly. To construct *pCS2+-sfGFP*, *sfGFP* coding sequence was amplified by PCR and cloned into *pCS2+* plasmid by Gibson assembly.

To construct *pCS2+-Nslmb-vhhGFP4*, *Nslmb-vhhGFP4* coding sequence was amplified from *pcDNA3-NSlmb-vhhGFP4* (Addgene plasmid #35579, (*Caussinus et al., 2011*)) by PCR and cloned into *pCS2+* plasmid by Gibson assembly.

To construct *pCS2+-OsTIR1-mCherry*, the *OsTIR1* and *mCherry* coding sequences were amplified from *pMK232-CMV-OsTIR1-PURO* (Addgene plasmid #72834, (*Natsume et al., 2016*)) and *pCS2+-lyn$_2$mCherry* (kind gift from Reinhard Köster and Scott Fraser) respectively, by PCR and cloned into *pCS2+* plasmid by Gibson assembly.

To construct *pCS2+-sfGFP-mAID*, the *sfGFP* coding sequence was amplified by PCR and the mAID tag sequence (*Morawska and Ulrich, 2013*) was generated by primer annealing and cloned into *pCS2+* plasmid by Gibson assembly.

To construct *pCS2+-fbxw11b-vhhGFP* (*pCS2+-zGrad*), the coding sequence for the N-terminal 217 amino acids of Fbxw11b was amplified by PCR from cDNA of 36 hpf embryos, fused to the *vhhGFP4* coding sequence and inserted into the *pCS2+* plasmid by Gibson assembly. This plasmid is available from addgene (plasmid #119716).

To construct *pCS2+-zif-1*, the coding sequence of *zif-1* was codon optimized for zebrafish by gene synthesis (IDT) and inserted into the *pCS2+* plasmid by Gibson assembly.

To construct *pCS2+-sfGFP-ZF1*, the coding sequence of *ZF1* was codon optimized for zebrafish by gene synthesis (IDT) and inserted together with *sfGFP* into the *pCS2+* plasmid by Gibson assembly.

The *pCS2+-vhhGFP4-hSPOP* and *pCS2+-vhhGFPmut-hSPOP* plasmids were previously described (*Shin et al., 2015*) and kindly gifted by Byung Joon Hwang.

## mRNA injection

Templates for in vitro mRNA transcription were generated by PCR or restriction digest of mRNA expression plasmids. mRNAs were transcribed using the mMESSAGE mMACHINE SP6 transcription Kit (Thermo Fisher Scientific). Injection mixes contained 50 ng/µl mRNAs except for the mAID experiment, where *OsTIR1-mCherry* mRNA was prepared as 10 ng/µl, 5 ng/µl and 1 ng/µl, with 0.1% Phenol Red Solution (LIFE TECHNOLOGIES). The injection mix was injected in one-cell-stage embryos. The quality of the mRNA was assessed post injection through gel electrophoresis using the remaining mix in the injection needle.

## Auxin-inducible degradation

Natural auxin 3-Indoleacetic acid (IAA, Sigma Aldrich) was dissolved in 100% ethanol at a concentration of 250 mM, protected from light and stored at −20°C. Injected embryos were raised in fish water (4 g/l instant ocean salt) without Methylene blue to avoid possible interference from the dye. 50 ng/µl *sfGFP-mAID* mRNA was co-injected with 10 ng/µl, 5 ng/µl or 1 ng/µl *OsTIR1-mCherry* mRNA. Injected embryos were dechorionated manually. At 8 hpf IAA was added to 500 µM final concentration (*Daniel et al., 2018*; *Zhang et al., 2015*). Images were taken on a Leica 165M FC Fluorescent Stereo Microscope equipped with a Leica DFC345 FX camera every hour for 3 hr.

## Image acquisition and quantification of mRNA-injected embryos

To quantify the ratio of sfGFP-ZF1 fluorescence to mScarlet-V5 fluorescence, injected embryos were dechorionated manually or by adding Pronase (Sigma Aldrich) to 0.3 mg/ml on petri dishes coated with 2% agarose in fish water at 9 hpf. Injected embryos were mounted in 0.5% low-melt agarose (National Diagnostics)/Ringer's solution (MgSO4 0.6 mM, CaCl2 1 mM, KCl 5 mM, NaCl 111 mM, HEPES 5 mM) on a slide. Mounted embryos were imaged on Leica SP5 II confocal microscope

equipped with HyD detectors (Leica Microsystems) using a Leica 20x (NA 0.7) objective and a Leica 40x water immersion lens (NA 1.1) in the case of Ab-SPOP-mediated GFP degradation.

All images were collected in the photon-counting mode with identical microscope settings. Quantification of signal intensity from injected embryos was performed using a custom-written ImageJ macro (*Source code 1*). Briefly, the macro selects a single Z-slice and generates a mask based on the mScarlet intensities using the Otsu thresholding algorithm. The mScarlet mask is then applied to the green and red channels of the same Z-slice to extract average GFP and mScarlet signal intensities only from the masked region. These procedures are repeated on the whole Z-stack. Average signal intensities from the first 40 Z-slices (62 µm) starting at the animal pole were used to calculate the green-to-red fluorescence intensity ratio of an embryo. The sfGFP-ZF1/mScarlet-V5 fluorescence intensity ratios for Ab-SPOP-mediated GFP degradation were calculated manually using ImageJ. Briefly, five nuclei per embryo were outlined and the average nuclear signal intensities in the green and red channels was extracted and used for the calculation of the green-to-red fluorescence intensity ratios in the nuclei. For quantification of signal intensities in the cytoplasm, a 20 × 20 pixels (3.79 µm x 3.79 µm) region was manually selected and the average signal intensities in the green and red channels were extracted to calculate the green-to-red fluorescence intensity ratio. The regions from five cells per embryo for five embryos were analyzed for each condition. The overview fluorescence images in *Figure 1D* and *Figure 1—figure supplement 1A* (AID) were collected on Leica 165M FC Fluorescent Stereo Microscope equipped with a Leica DFC345 FX camera.

## Heat-shock regimens

To determine the degradation kinetics of H2A-EGFP, *cxcr4b:h2a-EGFP; cxcr4b:h2a-mCherry* embryos with and without the *hsp70l:zGrad-IRES-h2a-TagBFP* transgene were heat shocked around 29–30 hpf at 39.5°C for 1 hr in a water bath, mounted in the 0.5% low-melt agarose/Ringer's solution with 0.4 mg/ml MS-222 anesthetic and imaged 80 min after the end of heat shock every 10 min for 9.5 hr on a Leica SP5 II confocal microscope equipped with HyD detectors (Leica Microsystems) using a 20x (NA 0.5) objective. Laser power was calibrated to 30 µW for 488 and 120 µW for 561. Pinhole was set to 85 µm.

To determine the degradation kinetics of Cdh1-sfGFP, *cdh1:cdh1-sfGFP; cdh1:cdh1-TagRFP* embryos with and without the *hsp70l:zGrad-IRES-h2a-TagBFP* transgene were mounted in the 0.5% low-melt agarose/Ringer's solution with 0.4 mg/ml MS-222 anesthetic, heat shocked around 31 hpf at 39.5°C for 30 min in a water bath and imaged 35 min after the end of heat shock every 10 min for 4.8 hr on a Leica SP5 II confocal microscope equipped with HyD detectors (Leica Microsystems) using a 40x (NA 0.8) objective. The laser power was calibrated to 32 µW for the 488 nm laser line and 115 µW for the 594 nm laser line. The pinhole was set to 230 µm.

To determine the degradation kinetics of Ctnna-Citrine, *ctnna:ctnna-Citrine/+* embryos with or without the *hsp70l:zGrad-IRES-h2a-TagBFP* transgene were mounted in the 0.5% low-melt agarose/ Ringer's solution with 0.4 mg/ml MS-222 anesthetic, heat shocked around 31 hpf at 39.5°C for 30 min in a water bath and imaged 35 min after the end of the heat shock every 20 min for 8 hr on a Leica SP8 confocal microscope equipped with HyD detectors (Leica Microsystems) using a 40x (NA 0.8) objective. The laser power was set to 31 µW for the 488 nm laser line. The pinhole was set to 106 µm. All imaged embryos were genotyped for *hsp70l:zGrad-IRES-h2a-TagBFP* by PCR.

To test the degradation of Sec-GFP by zGrad, *hsp70l:sec-GFP; hsp70:sec-mCherry* with and without the *hsp70l:zGrad-IRES-h2a-TagBFP* transgene were heat shocked around 24 hpf at 39.5°C for 1 hr in a water bath. Embryos were imaged at 30 hpf with a Leica 165M FC Fluorescent Stereo Microscope equipped with a Leica DFC345 FX camera. Imaged embryos were digested and genotyped for *hsp70l:zGrad-IRES-h2a-TagBFP* by PCR.

To observe the consequences of transient loss of Cxcr4b function, *cxcr4b:cxcr4b-EGFP-IRES-Kate2-CaaX-p7; cxcr4b -/-* embryos with and without the *hsp70l:zGrad-IRES-h2a-TagBFP* transgene were heat shocked at 31 hpf and 39.5°C for 1 hr in a water bath and imaged 30 min after the end of the heat shock every 20 min for 8 hr on a Leica SP8 confocal microscope equipped with HyD detectors (Leica Microsystems) using a 20x (NA 0.5) objective. The laser power was adjusted to 31 µW for the 488 nm laser line and to 73 µW for 594 nm laser line. The pinhole was set to 85 µm. Imaged embryos were genotyped for *hsp70l:zGrad-IRES-h2a-TagBFP* by PCR as described below.

## Analysis of the degradation kinetics after heat shock

Quantification of signal intensity from heat shocked embryos was performed using a custom-written ImageJ macro (*Source code 2*). For H2A-EGFP/H2A-mCherry ratio imaging, a region encompassing the primordium and the somites (100 × 50 pixels, 75.76 µm x 37.88 µm) was manually selected. The macro script duplicates the red channel, applies a Gaussian Blur (sigma = 1) and generates a mask based on the mCherry intensities using Renyi Entropy Thresholding algorithm for the primordium and the imageJ Default Thresholding for the somites. Then, the macro applies the mask on the green and red channels to extract H2A-EGFP and H2A-mCherry signal intensities only from the masked region. Then, the macro sum-projects the green and red channels and extracts mean intensities of H2A-EGFP and H2A-mCherry for each time point. Data from two independent imaging sessions were pooled for the analysis. Four embryos with *hsp70l:zGrad* transgene (two from each imaging session) and four embryos without *hsp70l:zGrad* transgene as control (two from each imaging session) were analyzed.

For Cdh1-sfGFP/Cdh1-TagRFP ratio imaging, first a region of 300 × 300 pixel (77.27 µm x 77.27 µm) at the center of the embryo was manually selected. The macro duplicates the red channel, applies a Gaussian Blur (sigma = 1) and generates a mask based on the TagRFP intensities using the imageJ Default Thresholding. Then, it applies the mask on the green and red channels to extract Cdh1-sfGFP and Cdh1-mCherry signal intensities only from the masked region. Then, the macro sum-projects the green and red channels and extracts the mean intensities of the Cdh1-sfGFP and Cdh1-TagRFP fluorescences for each time point. Three embryos with carrying the *hsp70l:zGrad* transgene and four control embryos without the *hsp70l:zGrad* transgene were analyzed.

For Ctnna-Citrine fluorescence intensity approximation, a region of 300 × 300 pixel (85.23 × 85.23 µm) at the center of the embryo was manually selected. The Z-stack comprising this region was maximum projected. The mean intensities of maximum-projected images were extracted over all time points. Four embryos with *hsp70l:zGrad* transgene and four embryos without *hsp70l:zGrad* transgene as control were analyzed.

For Cxcr4b-EGFP-to-Kate2-CaaX ratio imaging, the region of the primordium was manually selected and signal intensities were quantified using a custom-written ImageJ macro (*Source code 3*). Briefly, the macro script duplicates the red channel, applies a Gaussian Blur (sigma = 2), maximum-projects the Z-stack, generates a mask based on the Kate2 intensities using the imageJ Default Thresholding, fills holes, erodes and dilates once (erosion and dilation reduces noise). Then, the macro sum-projects the green and red channels and applies the mask to the green and red channels to extract Cxcr4b-EGFP and Kate2-CaaX signal intensities only from the masked regions for each time point. Five embryos with *hsp70l:zGrad* transgene and five embryos without *hsp70l:zGrad* transgene as control were analyzed.

## Analysis of the circularity of the primordium

To quantify the morphology of the primordium, we defined the extension of the primordium as the first 100 µm from the tip of the primordium. Using Fiji, the primordium region was manually cropped in the red channel based on the Kate2-CaaX fluorescence intensities. Then, a median filter (six pixels) was applied and the background was subtracted. Images were rendered binary using the Huang thresholding algorithm to obtain clear outlines of the primordium. Finally, we quantified the circularity of the primordium for each time point using the 'Analyze Particles' macro in Fiji. The circularity is defined as $circularity = 4pi(area/perimeter^2)$.

## Cumulative migration distance and kymographs analysis

To quantify the migration distance of the primordium over time, the 'Manual Tracking' plugin by Fabrice Cordelieres in Fiji was used to track the tip of the primordium. Kymographs were drawn using the 'KymoResliceWide' plugin by Eugene Katrukha and Laurie Young in Fiji with a width of 5 pixel.

Image acquisition and quantification of zGrad-mediated Cxcr4b-EGFP degradation with zGrad expressed from the *cxcr4b* promoter *cxcr4b:cxcr4b-EGFP-IRES-Kate2-CaaX-p1* embryos and *cxcr4b:cxcr4b-EGFP-IRES-Kate2-CaaX-p1; cxcrb4:zGrad-IRES-h2a-TagBFP* embryos were sorted for the expression of the transgenesis marker (the *cxcr4b:cxcr4b-EGFP-IRES-Kate2-CaaX-p1* transgene expresses dsRed in the lens and the *cxcrb4:zGrad-IRES-h2a-TagBFP* transgene expresses mScarlet in

the myocardium of the heart) at 29 hpf. Ten embryos of each genotype were mounted in 0.5% low-melt agarose/Ringer's solution with 0.4 mg/ml MS-222 anesthetic on a slide. Embryos were imaged at 33 to 34 hpf using a Leica SP8 confocal microscope equipped with HyD detectors (Leica Microsystems) using a 40x (NA 1.1) objective. The laser power was calibrated to 27 µW for the 488 nm laser line and 82 µW for the 594 nm laser line. The pinhole was set to 77.17 µm. To quantify the signal intensity ratio of EGFP/Kate2 in the primordium, the primordium was manually selected and fluorescent intensities for EGFP and Kate2 were extracted using the same custom-written ImageJ macro as described in the quantification of Cdh1-sfGFP degradation kinetics.

## Degradation of Ctnna-Citrine by zGrad mRNA injection

1–2 nl of 50 ng/µl of *zGrad* mRNA or 50 ng/µl of *sfGFP* mRNA were injected in one-cell stage zygotic, maternal and maternal zygotic *ctnna:ctnna-Citrine/ctnna:ctnna-Citrine* embryos. Possible degradation of the mRNA in the injection mix was assessed by electrophoresis of the injection mix in the needle after injection of the embryos. For imaging, the embryos were mounted in 0.5% low-melt agarose/Ringer's solution on a slide. Images of injected embryos were obtained using an Axioplan Microscope (Zeiss) equipped with an Axiocam (Zeiss) and a 10x (NA 0.5) objective for *Figure 3B and E* and a 5x (NA 0.25) objective for *Figure 3H*. The number of dead and alive embryos was scored at 24 hpf.

## Genotyping of *cxcr4b*$^{t26035}$ and *hsp70l:zGrad-IRES-H2A-TagBFP*

To distinguish endogenous *cxcr4b-/+* and *cxcr4b-/-* from the *cxcr4b:cxcr4b-EGFP-IRES-kate2-CaaX* transgene in *Figure 6* and *Figure 6—figure supplement 1*, the following outer and nested primer pairs were used: forward outer primer: GCAGACCTCCTGTTTGTCC reverse outer primer: CTAAG TGCACACATACACACATT forward nested primer: TCGAGCATGGGTACCATC reverse nested primer: CTTAATCATCCATGTGGAAAAG.

The reverse primers are designed to anneal to 3'UTR of the *cxcr4b* gene so that *cxcr4b-EGFP-IRES-Kate2-CaaX* transgene will not be amplified due to its large size. The PCR product was digested with the restriction enzyme HpyAV (NEB) to distinguish heterozygous and homozygous mutants.

To genotype *hsp70l:zGrad-IRES-h2a-TagBFP*, the region between *hsp70l* promoter and *fbxw11b* was amplified by PCR using the following outer and nested primer pairs: forward outer primer: TGAGCATAATAACCATAAATACTA reverse outer primer: ACCAGTTGGACTTGATCCATATG TCGACCACACCTCCAG forward nested primer: AGCAAATGTCCTAAATGAAT reverse nested primer: CAGAGGTGTTCATCTGCTC.

## Whole-mount in situ hybridization

The procedures for RNA probe synthesis and whole-mount in situ hybridization were done as previously described (*Thisse and Thisse, 2008*). The RNA probe against *cxcr4b* was previously described (*Knaut et al., 2003*). The template for the synthesis of the in situ RNA probe against *vhhGFP4* was amplified from *pcDNA3-NSlmb-vhhGFP4* (Addgene plasmid #35579) using the following primer pair: forward primer: ggccgtcgacATGATGAAAATGGAGACTGAC reverse primer: TAATACGAC TCACTATAGGGTTAGCTGGAGACGGTGACCTG.

The template for the synthesis of the in situ RNA probe against *EGFP* was amplified using the following primer pair: forward primer: GTGAGCAAGGGCGAGGAGCTG reverse primer: gaaatTAA TACGACTCACTATAgggCTTGTACAGCTCGTCCATGCC.

The template for the synthesis of the in situ RNA probe against *mCherry* was amplified using the following primer pair: forward primer: GTGAGCAAGGGCGAGGAGGACA reverse primer: gaaat-TAATACGACTCACTATAgggCTTGTACAGCTCGTCCATGCCGCCGG.

The RNA probe was synthesized using the Roche DIG labeling mix (Roche) and detected with an anti-DIG antibody coupled to alkaline phosphatase (1:5000, Roche) and NBT/BCIP stain (Roche). Embryos were mounted in the 3% Methyl cellulose (Sigma Aldrich). Images were collected on a Axioplan Microscope (Zeiss) equipped with an Axiocam (Zeiss) using a 10x (NA 0.5) objective.

## Quantification of the primordium migration distance

*cxcr4b:cxcr4b-EGFP-IRES-Kate2-CaaX-p1; cxcr4b-/+* fish were crossed to *cxcrb4:zGrad-IRES-h2a-TagBFP; cxcr4b-/-* fish and embryos were sorted based on the transgenesis markers (dsRed in lens for *cxcr4b:cxcr4b-EGFP-IRES-Kate2-CaaX-p1* and mScarlet in myocardium of heart for *cxcrb4:zGrad-IRES-h2a-TagBFP*) at 38 hpf and fixed in 4% PFA (Sigma Aldrich) in PBST overnight at room temperature. Whole-mount in situ hybridization against *cxcr4b* mRNA was performed as described above. Images for quantification of migration distance were taken in PBST on an Axioplan Microscope (Zeiss) equipped with an Axiocam (Zeiss) using a 10x (NA 0.5) objective. Both sides of the embryo were imaged and counted as individual replicates. Images for *Figure 6E* were taken by mounting embryos in 3% Methyl cellulose. *cxcr4b* mutant embryos were genotyped by PCR after image acquisition. The migrating distance of the primordium was quantified manually using ImageJ.

## Conditional Cdh1 loss-of-protein function through zGrad expression from a heat-shock promoter

The number of embryos from *cdh1:cdh1-sfGFP/+; cdh1+/-* with or without *hsp70l:zGrad* females crossed with *cdh1+/-* males was counted at 5 hpf. The dead embryos were removed at 24 hpf and embryos were sorted by Cdh1-sfGFP expression. At 25 hpf, embryos were heat shocked at 39.5℃ for 1 hr in a water bath. The embryos were mounted in 0.5% low-melt agarose/Ringer's solution on a plastic dish filled with Ringer's solution which was supplemented with 0.4 mg/ml MS-222 anesthetic. Images of embryos were recorded with an Axioplan Microscope (Zeiss) equipped with an Axiocam (Zeiss) and a 5x (NA 0.25) objective (panels B, C and D in *Figure 4*) and a 10x (NA 0.5) objective (panels C and D insets in *Figure 4*). The number of embryos with and without skin defects and lethality was scored at 32 hpf. Embryos were transferred into TRIzol reagent (Thermo Fisher Scientific) to extract mRNA. mRNA was reverse transcribed into cDNA with the Super Script III First-Strand Synthesis System kit (Thermo Fisher Scientific). To distinguish cDNA from the endogenous *cdh1* mRNA and the transgenic *cdh1:cdh1-sfGFP* mRNA, the following outer and nested primer pairs were used: forward outer primer: CAATATAACAGGCTCTGGGCAGAT reverse outer primer: ATAAAAGAGTCTCATATTTTGC forward nested primer: GTCAAGAATGCTTTGGATCG reverse nested primer: AATCATTGAGCCTTTTGCACC.

Due to the short elongation time, these PCRs only amplified the cDNA from the endogenous *cdh1* locus. The PCR amplicons were purified with the QIAquick PCR Purification Kit (QIAGEN) and sequenced with the primer TGTCCGTTATAGAGAGAAGC. Genotyping for *hsp70l:zGrad* transgenic embryos was also done from cDNA by PCR with the following primer pair: forward primer: ATGGATCAAGTCCAACTGG reverse primer: TTAGCTGGAGACGGTGACCC.

## DASPEI staining and neuromast position quantification

The location of neuromasts was assessed by staining 4 dpf live embryos with 25 ug/mL DASPEI (2-(4-(dimethylamino)styryl)-N-Ethylpyridinium Iodide, Invitrogen). Images were collected on Leica 165M FC Fluorescent Stereo Microscope equipped with a Leica DFC345 FX camera. The location of the last neuromast on the trunk was quantified as ratio between the length from the head to the last neuromast on the trunk divided by the length of the whole embryo (see *Figure 6—figure supplement 1B,C*).

## Statistical analysis

Statistical tests were performed using R and R-studio software. To compare two sample sets, first each sample set was tested using the Kolmogorov-Smirnov test to determine whether the sample set was normally distributed. Then, we tested whether the sample sets had the same standard deviation using the F-test. Based on the result of F-test, the two sample sets were compared by either the Welch's t-test or the Student's t-test (*Figures 1H*, *6D and F*, *Figure 6—figure supplement 1C*). To analyze the fold-change curves in *Figure 2C, E, H and K*, the curves were fitted to a one-exponential decay model (Y = Span*exp(-k*X)+Plateau) using Prism 7 (Graphpad). The values of T1/2 and the plateau, which we assumed to be the value of maximal degradation, were extracted from the fitted curves. In *Figure 5D* (inset, 30 min to 240 min), two data sets were compared by paired-t test using Prism.

## Acknowledgements

We thank S Lau for critical comments; D Kane, L Trinh and S Fraser for reagents, T Gerson for excellent fish care and Dorus Gadella (Addgene plasmid # 85042), Markus Affolter (Addgene plasmid # 35579), Masato Kanemaki (Addgene plasmid # 72835) and Byung Joon Hwang (Ab-SPOP and Abmut-SPOP) for plasmids. This work was supported by NIH grants HD088779 (HK), NS102322 (HK) and NYSTEM institutional training grant C322560GG (NY).

## Additional information

### Funding

| Funder | Grant reference number | Author |
|---|---|---|
| New York State Stem Cell Science | C322560GG | Naoya Yamaguchi |
| Eunice Kennedy Shriver National Institute of Child Health and Human Development | R21HD088779 | Holger Knaut |
| National Institute of Neurological Disorders and Stroke | R01NS102322 | Holger Knaut |

The funders had no role in study design, data collection and interpretation, or the decision to submit the work for publication.

### Author contributions

Naoya Yamaguchi, Conceptualization, Resources, Data curation, Software, Formal analysis, Funding acquisition, Validation, Investigation, Visualization, Methodology, Writing—original draft, Writing—review and editing; Tugba Colak-Champollion, Resources, Writing—review and editing; Holger Knaut, Conceptualization, Supervision, Funding acquisition, Writing—original draft, Project administration, Writing—review and editing

### Author ORCIDs

Naoya Yamaguchi http://orcid.org/0000-0003-0573-695X
Tugba Colak-Champollion http://orcid.org/0000-0001-5843-1717
Holger Knaut http://orcid.org/0000-0002-8399-8720

### Ethics

Animal experimentation: This study was performed in strict accordance with the recommendations in the Guide for the Care and Use of Laboratory Animals of the National Institutes of Health. All of the animals were handled according to approved institutional animal care and use committee (IACUC) protocols (IA16-00788_AMEND3) of the NYU School of Medicine.

### Decision letter and Author response

Decision letter https://doi.org/10.7554/eLife.43125.027
Author response https://doi.org/10.7554/eLife.43125.028

## Additional files

### Supplementary files

• Source code 1. ImageJ macro to analyze intensity of GFP and RFP channels of images from mRNA-injected embryos.
DOI: https://doi.org/10.7554/eLife.43125.022

• Source code 2. ImageJ macro to analyze signal intensity of the GFP and the RFP channel in time lapse movies from cxcr4b:h2a-EGFP; cxcr4b:h2a-mCherry embryos.
DOI: https://doi.org/10.7554/eLife.43125.023

• Source code 3. ImageJ macro to analyze signal intensity of the GFP and the RFP channel in time lapse movies from Cxcr4b-EGFP-to-Kate2-CaaX embryos.
DOI: https://doi.org/10.7554/eLife.43125.024

• Transparent reporting form
DOI: https://doi.org/10.7554/eLife.43125.025

### Data availability

All data generated or analysed during this study are included in the manuscript and supporting files. Two main tables are provided: Table 1 as a detail numerical data set for supporting Cdh1 transgene is fully functional; Table 2 as detail numerical data set for protein degradation kinetics exported from fitting against the one-exponential decay model. Source code files 1-3 describe the custom Image J macros written to analyze data. The pCS2-zGrad plasmid is available from Addgene (https://www.addgene.org/119716/). Transgenic fish lines are available from our lab upon request to the corresponding author.

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
