## [Decision Letter]

Thank you for submitting your article "zGrad is a nanobody-based degron system that inactivates proteins in zebrafish" for consideration by *eLife*. Your article has been reviewed by three peer reviewers, and the evaluation has been overseen by a Reviewing Editor and Didier Stainier as the Senior Editor. The following individual involved in review of your submission has agreed to reveal his identity: Thomas F Schilling (Reviewer #2).

The reviewers have discussed the reviews with one another and the Reviewing Editor has drafted this decision to help you prepare a revised submission. There was broad agreement that your work would be suitable for *eLife* after the points raised below are addressed.

*Reviewer #1:*

The manuscript submitted by Holger Knaut and colleagues for publication in *eLife* describes a novel approach to study protein function in zebrafish, a model system that has made major contribution to a better understanding of vertebrate development due to its ease to do live imaging studies at cellular resolution.

Upon testing a number of different degradation systems, the authors adapted a previously described method to degrade GFP-fusion proteins with a nanobody recognizing GFP, fused itself to a F-box of the *Drosophila* protein Slmb, targeting the GFP-fusion protein to the proteasome. Since the authors found that the *Drosophila* F-box did not lead to efficient degradation, they exchanged the latter with the F-box of the corresponding gene from zebrafish, and then showed that this construct is capable to degrade nuclear, transmembrane and cytoplasmic FP-tagged proteins. Most importantly, using endogenously tagged FP proteins, the authors showed that zGrad was capable to generate loss of function phenotypes in the zebrafish, opening the door to degrade proteins in a time- and tissue-dependent manner in unprecedented fashion. More importantly, protein degradation allows removing maternal contribution of the protein of interest (if a FP tagged rescue or endogenous version is available); this is particularly important for cell biological studies, since many widely expressed proteins are also maternally contributed.

The study is done very carefully and in a quantitative manner. This study was only possible since the authors had functional FP constructs of their genes of interest.

*Reviewer #2:*

In this paper the authors report their development of a nanobody-based method for targeted protein degradation in zebrafish. They show that by adapting a deGradFP system used in *Drosophila* they can obtain efficient degradation of nuclear, cytoplasmic and transmembrane proteins in both a temporally- and spatially-controlled manner, while other similar degradation systems are less effective. They confirm that using this zGrad system they can obtain previously published loss-of-function phenotypes for E-cadherin, E-catenin and Cxcr4b. They then go beyond previous mutant studies of these factors to address temporal requirements – such as maternal versus zygotic functions of E-catenin, and Cxcr4b in collective migration of the lateral line primordium – as well as tissue-specific promoters to disrupt Cxcr4b in specific cell types. This is a major technical advance, as tools for conditional loss-of-function approaches in zebrafish are desperately needed. The experiments are well designed, rigorously analyzed and the results are convincing. It is somewhat less innovative conceptually given the fact that a similar system exists in flies and functions of all of the genes examined here have been studied before. In addition, there are some concerns about variability in the effectiveness of zGrad and its reproducibility.

1) The paper shows a wide variety of efficiencies in the reduction in levels as well as the timing of degradation for different zGrad target proteins. As proof of principle with sfGFP-ZF1 they co-inject one-cell stage embryos and assay protein levels 9 hours later (Figure 1). What are the results if assayed at 2-3 hpf? The authors argue that the system should work within that time frame. For H2A-EGFP the onset of degradation is estimated as 140-200 min while for Ctnna-Citrine it appears to be as early as 65 min. How do the authors account for these differences in onset?

2) Experiments with H2A-EGFP show widely varied reductions in protein levels in different tissues, which the authors argue reflects different levels of endogenous H2A production. This raises the question of how functional experiments with proteins for which function is unknown can deal with the potential differences in efficiency introduced by variation in the synthesis of new protein? Negative results obtained with zGrad will be extremely hard to interpret. For Figure 2D-E the authors should confirm the hypothesis that the difference in degradation efficiency between somites and lateral line primordia is due to different expression levels of H2A-GFP with in situs for eGFP and mCherry.

3) Variability in the effectiveness of zGrad for Cdh1 to cause cell adhesion defects is explained as being due to variability in zGrad levels. However again this complicates interpretation. Can zGrad levels be monitored and correlated with variability in phenotypic defects? The authors should genotype the embryos to identify those that are *cadh+/-;cdh1:cd1-sfgfp;hsp70l:zGrad* as the phenotype can only be visualized in these embryos.

4) Interpretation of the results also depends to a great extent on what is co-injected in a given experiment, in terms of amount, and how long it persists in the embryo. This is particularly a potential problem in experiments addressing maternal versus zygotic function.

5) The authors should compare the different deGradFP systems used in zebrafish more clearly either in the Introduction or Discussion section to help the reader appreciate the improvements in their system over the previous ones. Since the authors start with results from existing methods and show them in figure supplements, it would be helpful if they present their general experimental strategy including the model for the degron system in the first supplementary figure.

6) Clarify the choice of the Slimb system over the *C. elegans* system, considering that both had similar efficiency of knockdown (19% and 17%).

7) Clarify why the results shown in Figure 2—video 1 and Figure 5—video 1 are so similar – slight delays in lateral line migration in zGrad expression embryos, which recover over time – considering the difference in their backgrounds?

8) Figure 3. The *ctnna*-citrine strategy is unclear. Are the authors using *ctnna* heterozygotes for the cross? What is zGrad targeting in this cross, since the embryos appear to be expressing a citrine fusion protein? Does vhhGFP4 recognize citrine?

*Reviewer #3:*

In the manuscript "zGrad is a nanobody-based degron system that inactivates proteins in zebrafish” Yamaguchi, Colak-Champollion and Knaut report the establishment of a nanobody-based tool to degrade GFP-tagged proteins in zebrafish. The authors developed a simple, efficient and spatially- and temporally inducible 1-component system that targets GFP-tagged proteins of interest with different subcellular localizations for degradation by the endogenous ubiquitin-based machinery. Degradation is sufficiently efficient to result in a) loss-of-function phenotypes after global induction of protein degradation, and b) hypomorphic and/or more restricted phenotypes for tissue/time-specific induction of protein degradation.

Compared to the recently published mAID nanobody based 2-component system for zebrafish by the Mansfeld/Norden labs (Daniel et al., 2018), which is most similar to the zGrad tool presented here, zGrad provides a valuable and potentially superior drug-independent 1-component alternative. The main difference of the two systems is the Auxin/TIR1-dependence (mAID) versus independence (zGrad) by using a different degradation-inducing fusion protein to the vhh-GFP4 nanobody: In the case of the published mAID system (Daniel et al., 2018), the nanobody vhh-GFP4 is fused to AID to link via the adaptor protein TIR1 and Auxin to the SCF degradation complex. Degradation in this system was reported to depend on the presence of Auxin. In case of zGrad, the nanobody vhh-GFP4 is fused to Fbxw11b, which binds directly in an Auxin-independent manner to the SCF-complex member Skp1 and thus does not need an additional adaptor to link to the SCF degradation complex. The authors show that the system is inducible by hs- or tissue-specific control of the nanobody fusion construct. Thus, no drugs are needed to induce degradation of GFP-tagged proteins of interest and the newly developed system presented here. Most importantly, the authors present convincing data that zGrad is efficient enough to cause protein-depletion at a level that results in loss-of-function phenotypes.

Overall, zGrad provides a very elegant, simple system to induce protein degradation without the need of drugs or additional adaptor proteins, which also circumvents the problem of potential leakiness of directly AID-tagged target proteins (degradation even in the absence of Auxin). zGrad presents a valuable contribution for functional studies in zebrafish, and will be an important, versatile alternative (and potentially superior, depending on the application) tool that nicely complements other available tools for protein degradation.

The data is of very high quality, and the experiments support the conclusions of the paper.

I have no major concerns.

---

## [Author Response]

Reviewer #2:1) The paper shows a wide variety of efficiencies in the reduction in levels as well as the timing of degradation for different zGrad target proteins. As proof of principle with sfGFP-ZF1 they co-inject one-cell stage embryos and assay protein levels 9 hours later (Figure 1). What are the results if assayed at 2-3 hpf?

We repeated the mRNA injection experiments and observed sfGFP-ZF1 protein degradation as early as 2.5 hpf. Although the GFP signal in the controls was dim at 2.5 hpf, it was clearly higher than the GFP signal in embryos co-injected with zGrad mRNA (Figure 1—figure supplement 3).

The authors argue that the system should work within that time frame. For H2A-EGFP the onset of degradation is estimated as 140-200 min while for Ctnna-Citrine it appears to be as early as 65 min. How do the authors account for these differences in onset?

We do not know why Cdh1-GFP and Ctnna-Citrine are degraded faster than H2A-GFP. One difference between these three proteins is their subcellular localization. H2A-GFP is nuclear while Cdh1-GFP is a transmembrane protein with a cytoplasmic tail and Ctnna-Citrine resides fully in the cytoplasm. It is possible that nuclear proteins are less accessible to zGrad-mediated degradation than proteins which are partly of wholly residing in the cytoplasm. Also, the nature of the protein, the complex in which the protein resides and the number of potential ubiquitination sites in the protein will likely contribute to rate of zGrad-mediated degradation.

2) Experiments with H2A-EGFP show widely varied reductions in protein levels in different tissues, which the authors argue reflects different levels of endogenous H2A production. This raises the question of how functional experiments with proteins for which function is unknown can deal with the potential differences in efficiency introduced by variation in the synthesis of new protein? Negative results obtained with zGrad will be extremely hard to interpret. For Figure 2D-E the authors should confirm the hypothesis that the difference in degradation efficiency between somites and lateral line primordia is due to different expression levels of H2A-GFP with in situs for eGFP and mCherry.

We agree that this is a caveat when assessing protein function through protein depletion. However, since the protein of interest will be tagged with a fluorescent protein (FP) it is possible to assess the degree of protein depletion in different tissues and use this information when drawing conclusions. Also, if this is a problem increasing the expression levels of zGrad is an option to increase the degree of FP-tagged protein degradation.

To confirm that the differences in H2A-EGFP degradation efficiencies are due to difference in H2A-EGFP expression levels in somites and the posterior lateral line primodium, we stained embryos for *EGFP* and *mCherry* mRNA at 13 hpf (8-somite stage) and at 33 hpf. Consistent with a previous study [1], we find that the *cxcr4b* promoter drives *EGFP* and *mCherry* mRNA expression in the somites at 13 hpf but not at 33 hpf anymore. In contrast, *cxcr4b* drives expression of *EGFP* and *mCherry* mRNA in the primordium at 33 hpf (Figure 2—figure supplement 1). This indicates that the expression of H2A-EGFP protein in the somites at around 30 hpf is due to H2A-EGFP protein perdurance from past *cxcr4b* promoter activity.

3) Variability in the effectiveness of zGrad for Cdh1 to cause cell adhesion defects is explained as being due to variability in zGrad levels. However again this complicates interpretation. Can zGrad levels be monitored and correlated with variability in phenotypic defects? The authors should genotype the embryos to identify those that are cadh+/-;cdh1:cd1-sfgfp;hsp70l:zGrad as the phenotype can only be visualized in these embryos.

We agree and now included an experiment in which we genotyped the embryos for *hsp70l:zGrad* and *cdh1* using cDNA isolated from individual embryos which were phenotypically wild-type or displayed detaching skin cells. As shown in Figure 4, there is a perfect correlation between the heat-shocked embryos with a skin defect and the genotype (*cdh1-/-; cdh1:cdh1-sfGFP; hsp70l:zGrad*, n = 7) and heat-shocked embryos without a skin defect and the genotype (*cdh1-/-; cdh1:cdh1-sfGFP*, n = 1 and *cdh1*-/+ or *cdh1*+/+; *cdh1:cdh1-sfGFP; hsp70l:zGrad* n = 8). For genotyping, we could not use genomic DNA because the *cdh1:cdh1-sfGFP* transgene spans the *cdh1* locus and cannot be distinguished from the endogenous *cdh1* locus by PCR-based genotyping approaches. However, the mRNAs for *cdh1* and *cdh1-sGFP* can be distinguished by PCR based on the different amplicon sizes. We therefore developed a cDNA-based genotyping protocol that we included in our revised manuscript.

4) Interpretation of the results also depends to a great extent on what is co-injected in a given experiment, in terms of amount, and how long it persists in the embryo. This is particularly a potential problem in experiments addressing maternal versus zygotic function.

We agree that the levels and turnover of zGrad will greatly affect the depletion of the FP-tagged target protein. We would like to note, however, that the degree of target protein degradation can easily be monitored by following the fluorescence from the FP-tag. Experiments addressing the function of maternal versus zygotic proteins can be designed in such a way that only the maternal protein or only the zygotic protein is subject to zGrad-mediated degradation. By crossing females homozygous for the gene encoding the FP-tagged target protein to wild-type males homozygous for the untagged/wild-type gene, one can generate embryos in which only the maternal target protein will be degraded by zGrad. Conversely, by crossing females heterozygous for the gene encoding the FP-tagged target protein to males of the same genotype, one can generate embryos in which only the zygotic target protein will be degraded by zGrad. However, as noted by Reviewer 2, in both cases one caveat is that zGrad may not completely deplete the tagged target protein. Therefore, monitoring the fluorescence of the target protein will be essential to assess the degree of protein depletion.

5) The authors should compare the different deGradFP systems used in zebrafish more clearly either in the Introduction or Discussion section to help the reader appreciate the improvements in their system over the previous ones. Since the authors start with results from existing methods and show them in Supplement Figures, it would be helpful if they present their general experimental strategy including the model for the degron system in the first Supplementary Figure.

We agree and now included a discussion of the different degron-based protein depletion systems. We also included schematic cartoons of the different degron-based protein depletion systems (Figure 1A).

6) Clarify the choice of the Slimb system over the C. elegans system, considering that both had similar efficiency of knockdown (19% and 17%).

We tried to adapt the ZIF-1 system to zebrafish by replacing the SOCS box motif in ZIF-1 with a SOCS box from zebrafish. We chose the SOCS box from the zebrafish d-Asb11 protein because it has been shown that d-Asb11 associates with ECS (Elongin BC-Cul2/Cul5-SOCS-box protein) ubiquitin ligase complex to degrade DeltaA in zebrafish [2]. However, this ZIF1-d-Asb11 fusion did not degrade GFP-ZF1 more efficiently than ZIF1 (Author response image 1). This is why we focused on zGrad for our further experiments

**Author response image 1. respfig1:** A zebrafish SOCS box does not increase ZIF-1-mediated degradation of ZF1-tagged GFP. (**A**) Schematics of the chimeric protein design carrying ZF1-binding motif from ZIF-1 and SOCS box motif from d-Asb11. (**B**) Embryos injected with sfGFP-ZF1 mRNA or sfGFP-ZF1 and ZIF-1-d-Asb11 mRNA.

7) Clarify why the results shown in Figure 2—video 1 and Figure 5—video 1 are so similar – slight delays in lateral line migration in zGrad expression embryos, which recover over time – considering the difference in their backgrounds?

Figure 2—video 1 documents the degradation of H2A-GFP upon expression of zGrad from the *cxcr4b* promoter. We used these time lapse videos to assess the degradation efficiency and kinetics of H2A-EGFP in the nuclei of the primordium. The embryos are wild type and express *cxcr4b* from the endogenous locus, and we did not see any migration defects upon zGrad expression.

Figure 5—video 1 documents the consequences of transient Cxcr4b protein loss of function on primordium migration. We used time lapse videos to assess how the depletion of Cxcr4b-EGFP upon zGrad expression from a heat shock promoter in *cxcr4b* mutant embryos affects the migration of the primordium. Note that in *cxcr4b* mutant embryos that carry the *cxcr4b:cxcr4b-EGFP-IRES-Kate2-CaaX* transgene the migration of the primordium is restored [3]. We found that transient expression of zGrad efficiently depleted Cxcr4b-EGFP and resulted in primordium stalling. Once Cxcr4b-EGFP levels recovered the primordium resumed its migration. Thus, the two videos document two different aspects – zGrad-mediated depletion of H2A-EGFP and zGrad mediated Cxcr4b loss of protein function.

8) Figure 3. The ctnna-citrine strategy is unclear. Are the authors using ctnna heterozygotes for the cross? What is zGrad targeting in this cross, since the embryos appear to be expressing a citrine fusion protein? Does vhhGFP4 recognize citrine?

The *Gt(ctnna-citrine)Ct3a* allele is a gene-trap line, in which the coding sequence for the GFP-variant Citrine [4] hopped into the *ctnna* locus to generate a Ctnna-Citrine functional fusion protein. The gene-trap coding for *citrine* is inserted in the intron between exon 7 and exon 8 of *ctnna* [5]. We found that Ctnna-Citrine protein is degraded by zGrad (Figure 2I-K) indicating that zGrad recognizes Citrine. This is consistent with the observation that deGradFP recognizes and depletes EYFP-tagged Histone in flies [6].

We generated embryos in which only maternal Ctnna protein was tagged with Citrine, only zygotic Ctnna protein was tagged with Citrine or both maternal and zygotic Ctnna protein was tagged with Citrine. We injected such embryos with *zGrad* mRNA to assess whether zGrad depletes maternal, zygotic and maternal and zygotic proteins and induces loss of protein function phenotypes. For these genotypes our nomenclature might have been confusing. We indicated fish homozygous for *ctnna:ctnna-citrine* simply as *ctnna:ctnna-citrine* fish instead of *ctnna:ctnna-citrine/ctnna:ctnna-citrine* fish. We are sorry for the confusion this might have caused and now indicate the full genotype of the embryos and adult fish.

[1] Chong SW, Emelyanov A, Gong Z, Korzh V. Expression pattern of two zebrafish genes, cxcr4a and cxcr4b. Mech Dev 2001;109:347–54.

[2] Diks SH, Sartori da Silva MA, Hillebrands J-L, Bink RJ, Versteeg HH, van Rooijen C, et al. d-Asb11 is an essential mediator of canonical Delta–Notch signalling. Nature Cell Biology 2008;10:1190–8.

[3] Venkiteswaran G, Lewellis SW, Wang J, Reynolds E, Nicholson C, Knaut H. Generation and Dynamics of an Endogenous, Self-Generated Signaling Gradient across a Migrating Tissue. Cell 2013;155:674–87.

[4] Griesbeck O, Baird GS, Campbell RE, Zacharias DA, Tsien RY. Reducing the environmental sensitivity of yellow fluorescent protein. Mechanism and applications. J Biol Chem 2001;276:29188–94.

[5] Trinh LA, Hochgreb T, Graham M, Wu D, Ruf-Zamojski F, Jayasena CS, et al. A versatile gene trap to visualize and interrogate the function of the vertebrate proteome. Genes Dev 2011;25:2306–20.

[6] Caussinus E, Kanca O, Affolter M. Fluorescent fusion protein knockout mediated by anti-GFP nanobody. Nat Struct Mol Biol 2011;19:117–21.